# The Fluorescent Veil: A Stealthy and Effective Physical Adversarial Patch Against Traffic Sign Recognition

**Shuai Yuan**[1], **Xingshuo Han**[2], **Hongwei Li**[1], **Guowen Xu**[1]\*,
**Wenbo Jiang**[1], **Tao Ni**[3,4], **Qingchuan Zhao**[3], **Yuguang Fang**[3]

[1]University of Electronic Science and Technology of China, Chengdu, China
[2] Nanjing University of Aeronautics and Astronautics, Nanjing, China
[3] City University of Hong Kong, Hong Kong, China
[4] King Abdullah University of Science and Technology, Thuwal, Saudi Arabia
mk2456mk@gmail.com, xingshuo.han@nuaa.edu.cn, tao.ni@kaust.edu.sa,
{hongweili, guowen.xu, wenbo_jiang}@uestc.edu.cn,
{cs.qczhao, my.Fang}@cityu.edu.hk,

## Abstract

Recently, traffic sign recognition (TSR) systems have become a prominent target for physical adversarial attacks. These attacks typically rely on conspicuous stickers and projections, or using invisible light and acoustic signals that can be easily blocked. In this paper, we introduce a novel attack medium, i.e., *fluorescent ink*, to design a stealthy and effective physical adversarial patch, namely FIPatch, to advance the state-of-the-art. Specifically, we first model the fluorescence effect in the digital domain to identify the optimal attack settings, which guide the real-world fluorescence parameters. By applying a carefully designed fluorescence perturbation to the target sign, the attacker can later trigger a fluorescent effect using invisible ultraviolet light, causing the TSR system to misclassify the sign and potentially leading to traffic accidents. We conducted a comprehensive evaluation to investigate the effectiveness of FIPatch, which shows a success rate of $98.31\%$ in low-light conditions. Furthermore, our attack successfully bypasses five popular defenses and achieves a success rate of $96.72\%$.

## 1 Introduction

Traffic sign recognition (TSR) plays a pivotal role in autonomous driving by visually detecting and classifying traffic signs to ensure driving safety under various road situations. However, most TSR systems were built atop machine-learning models that are inherently suspected and also shown to be subject to adversarial attacks Ilyas et al. [2019], Zhang et al. [2020], making TSR systems work incorrectly. In particular, these attacks were launched by using adversarial examples (AEs) to introduce subtle perturbations to normal images, and these perturbations could deceive the underlying machine-learning model into making incorrect detection and classification. Numerous efforts have been devoted to investigating AEs in TSR systems, and recent focus has shifted from the digital domain Goodfellow et al. [2014], Carlini and Wagner [2017], Su et al. [2019] to the physical domain Tu et al. [2020], Sayles et al. [2021], Yan et al. [2022], Wang et al. [2023], Sato et al. [2024]. These adversarial attacks target either cameras or traffic signs. Since attacks on cameras involve the impractical requirement of physically accessing the target vehicle, we focus on adversarial attacks targeting traffic signs.

Existing physical AEs against TSR exploits stickers, light and acoustic signals. Specifically, extensive studies have been conducted on sticker-based adversarial attacks Eykholt et al. [2018], Song et al.

---

\*Corresponding author. E-mail: guowen.xu@uestc.edu.cn

39th Conference on Neural Information Processing Systems (NeurIPS 2025).

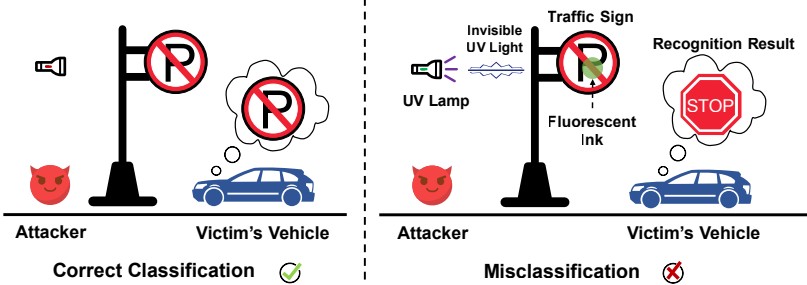

Figure 1: An example of FIPatch attack. The attacker first applies carefully crafted fluorescent ink to a traffic sign. When the attack is not triggered, the TSR system correctly classifies the sign. However, the attacker can uses a UV lamp from the side of the road to launch the attack against the victim's vehicle, causing it to misrecognize a no-parking sign as a STOP sign, which leads to the vehicle suddenly stopping.

[2018], Liu et al. [2019], which are easy to deploy with low cost. Note that we consider printed perturbations Ye et al. [2021], Jia et al. [2022], Wei et al. [2022] as stickers as well. However, these adversarial stickers are visually suspicious and attack all vehicles indiscriminately once deployed. Furthermore, some works used visible light, such as projections Lovisotto et al. [2021] and lasers Duan et al. [2021], to construct physical AEs capable of flexibly triggering attacks. Unfortunately, such visible light can be easily tracked, revealing the location of the attacker. Recently, Sato et al. Sato et al. [2024] constructed physical AEs to mislead TSR via an infrared (IR) laser, which is invisible to human eyes. However, IR-based attacks are vulnerable to simple physical defenses such as IR filters. Unlike others, Zhu et al. Zhu et al. [2023] designed adversarial patches triggered by acoustic signals, which can be easily blocked by physical signal protection mechanisms Sharma et al. [2017], Lou et al. [2021]. Overall, existing physical AEs are either poorly stealthy or easily countered.

In this work, surprisingly, we discovered a new attack vector, i.e., *fluorescent ink*, that can significantly address the aforementioned issues and advance the state-of-the-art. Specifically, fluorescent ink is transparent in normal environments, making them "unnoticeable". On the other hand, fluorescent ink exhibits fluorescent effects after absorbing specific wavelengths of light, e.g., invisible ultraviolet (UV) light, which allows adversarial attacks to not only be flexibly triggered but also to bypass any filters. This is because the light emitted by fluorescent materials falls within the visible light spectrum NASA, meaning only visible-light filters could potentially block it. Nonetheless, no cameras in current AVs use visible-light filters, as such filters would significantly damage the camera's utility. Moreover, filtering a single color does not prevent the effectiveness of fluorescent ink in other colors. Although fluorescent ink demonstrates strong stealthiness and resistance to filters, designing an effective and robust physical adversarial patch using fluorescent ink remains non-trivial with the following challenges. First, it is challenging to simulate fluorescent effects and determine the optimal choices of massive factors in fluorescent ink, e.g., color, transparency, and size, for achieving high attack effectiveness. Second, fluorescent effects are easily influenced by real-world environments, including surroundings, ambient light, vehicle distance, and speed. Feasibility study is provided in Appendix A.

To address the above challenges, we design a stealthy and effective physical adversarial patch with fluorescent ink (FIPatch[2]), which leverages the aforementioned fluorescent properties. At a high level, an attacker applies carefully designed fluorescent ink to a target sign and later triggers a fluorescent effect using invisible UV light. The resulting fluorescent perturbations cause the TSR system to misclassify the sign, which could potentially lead to traffic accidents. Figure 1 shows an example of our FIPatch. In more detail, our methodology consists of four key modules. First, we develop a color-edge fusion method to automatically locate traffic signs, enabling precise application of fluorescent ink to the signs themselves, rather than invalid backgrounds. Second, to effectively simulate fluorescent effects, we model fluorescent perturbations on traffic signs by defining the various critical parameters of fluorescent ink, including colors, intensities, and perturbation sizes. Third, we design goal-based and patch-aware loss functions to achieve high attack success rates with minimal perturbations, supporting three attack goals: hiding attack, generative attack, and

---

[2]Our demonstration videos of FIPatch can be found at https://sites.google.com/view/fipatch-attack/home/.

misrecognition attack. Finally, to improve the robustness of FIPatch in the physical world, we present several fluorescence-specific transformation methods that simulate fluorescence perturbations for real-world attacks.

We perform extensive experiments using 10 TSR models to validate our attacks in both digital and physical settings. The evaluation results show that under low-light conditions, the success rates for both generative and misrecognition attacks are above $98.31\%$, while the success rate for hiding attacks is at least $87.81\%$. Additionally, we conduct ablation studies to examine the impact of various factors, such as color, size, and shape, and test how real-world environments e.g., distance, ambient light, and vehicle speed affect the robustness of FIPatch. We further evaluate the effectiveness of FIPatch in two specific attack scenarios. It is worth noting that we test 5 common defenses and find that FIPatch can achieve an attack success rate of at least $96.72\%$.

Our contributions are summarized as follows.

- We are the first to introduce *fluorescent ink* to construct physical adversarial patches.
- We design FIPatch, a systematic attack optimization framework tailored to the physical properties of fluorescent ink that achieves high stealthiness, effectiveness, robustness with low-cost.
- We extensively evaluate FIPatch for three attack goals in both digital and physical worlds against five popular defenses.

## 2 Background and Related Works

### 2.1 Traffic sign recognition

Generally, the TSR system is divided into two main steps: detection and classification. We briefly describe the popular detectors and classifiers. First, Yolov3 Redmon and Farhadi [2018] and Yolov5 Redmon et al. [2016] are classical one-stage detectors that achieve accurate object detection by dividing the image into grids and predicting both bounding boxes and categories. Other popular one-stage detectors include SSD Liu et al. [2016], RetinaNet Lin et al. [2017], and EfficientNet Tan and Le [2019]. In contrast, Faster R-CNN Ren et al. [2015] is one of the most popular two-stage detectors. Faster R-CNN first screens high-quality candidate target regions using a region proposition network, and then performs target classification and localization via a convolutional neural network. Some recent works such as HyperNet Kong et al. [2016], R-FCN Dai et al. [2016], Mask R-CNN He et al. [2017], and Cascade R-CNN Cai and Vasconcelos [2018] have also improved the performance of Faster R-CNN. Second, the classifiers usually receive images and a series of bounding boxes of traffic signs as input and then output the classification results of these traffic signs. Models such as VGG Simonyan and Zisserman [2014], GoogleNet Szegedy et al. [2015], ResNet He et al. [2016], and MobileNet Howard et al. [2017] are widely used classifiers in TSR systems.

### 2.2 Physical adversarial examples

Physical AEs are designed to deceive machine learning models by introducing perceptible perturbations to physical systems. In TSR, such attacks mainly target traffic signs or cameras, though camera-based attacks typically involve color or translucent films Li et al. [2019], Zolfi et al. [2021], Hu and Shi [2022]. Following most prior work, we focus on attacks against traffic signs.

The current attack mediums for traffic signs can be categorized into stickers, light signals, and acoustic signals. Specifically, some researchers Eykholt et al. [2018], Song et al. [2018], Liu et al. [2019], Wei et al. [2022] have used stickers to create physical AEs on traffic signs. Other studies Chen et al. [2019], Yang et al. [2020] designed full-size printable adversarial signs, which we also classify as stickers. However, once deployed, these stickers launch attacks continuously and indiscriminately.

To address this issue, some studies have explored the light to deceive TSR systems. The first type utilizes visible light (wavelengths from $400\,\mathrm{nm}$ to $800\,\mathrm{nm}$), implemented via projectors Lovisotto et al. [2021] or lasers Duan et al. [2021]. However, these light sources are easily tracked, which exposes the attacker. Moreover, the projector used in Lovisotto et al. [2021] is expensive, costing between \$1500 and \$44379. Recently, Sato et al. [2024] proposed an IR laser reflection (ILR) attack to mislead AV perception modules, which is invisible to human eyes. However, as highlighted by Sato

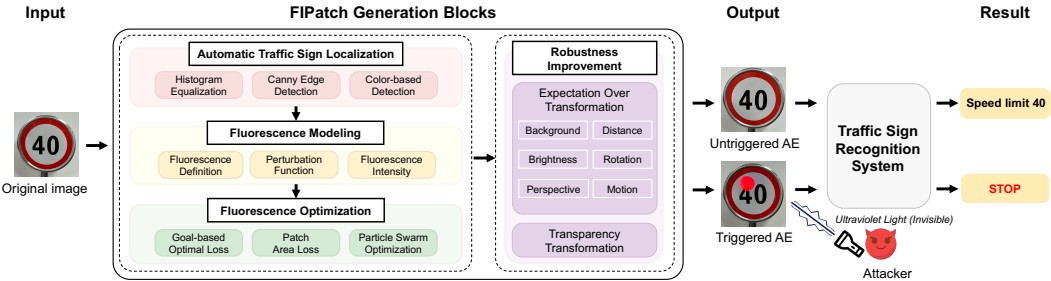

Figure 2: The workflow of our FIPatch.

et al. [2024], this attack only affects sensors without IR filters, indicating that IR filters can effectively block ILR attacks. dditionally, Zhu et al. Zhu et al. [2023] designed a physical adversarial patch triggered by acoustic signals. However, this method is impractical due to the challenges of using ultrasonic devices and can be easily countered by physical signal protection mechanisms Sharma et al. [2017], Lou et al. [2021]. In this paper, we introduce a novel attack medium that addresses the limitations of the aforementioned methods and may pose a significant threat to TSR systems.

## 3    Threat Model

### 3.1    Attack goals

We analyze TSR outputs and propose three attacks: hiding and generative attacks for detection, and a misrecognition attack for classification, defined as follows.

- **Hiding attack.** An attacker hides a traffic sign from the TSR system to cause detection failure.
- **Generative attack.** An attacker causes a TSR system to detect a forged traffic sign.
- **Misrecognition attack.** An attacker causes a TSR system to misclassify a traffic sign.

### 3.2    Attacker capabilities

In this work, we assume that the attack is black-box based, i.e., the attacker does not have direct access to the internal details of the target model, such as its architecture, parameters, or gradients. We assume the attacker has the following capabilities.

- **Direct access to traffic signs.** An attacker can physically access traffic signs.
- **No direct access to a victim's vehicle.** An attacker does not have digital or physical access to the victim's vehicle before or during any phase of an attack.
- **Launching an attack.** An attacker can launch an attack by either placing a UV lamp by the roadside and controlling it remotely to target the traffic sign, or by driving close to the victim's vehicle and using a UV lamp to target the traffic sign.

## 4    Methodology

To implement FIPatch in the physical world, it is essential to overcome the following challenges:

**Challenge 1:** How to accurately model fluorescent ink and determine the most effective attack parameters for FIPatch?

**Challenge 2:** How to enhance the robustness of FIPatch by leveraging the properties of fluorescent ink, making it more viable for real-world application?

To address these challenges, we propose a four-module FIPatch attack framework, as illustrated in Figure 2. The **Automatic Traffic Sign Localization** module automatically detects the valid region on a traffic sign to ensure consistency between the location of the perturbation in the digital and physical environments. The **Fluorescence Modeling** module defines and systematically models several key parameters related to fluorescent ink, including position, radius, color, and intensity. To the best of our knowledge, this is the first attempt to parameterize fluorescent perturbations for attack

optimization. The **Fluorescence Optimization** module optimizes these parameters using goal-based and patch-aware loss functions and employs a particle swarm optimization algorithm to identify the most effective attack configuration. These three modules collectively address Challenge 1.

To tackle Challenge 2, the **Robustness Improvement** module customizes multiple transformation distributions based on the properties of fluorescent materials to enhance the real-world robustness of FIPatch. The following subsections provide a detailed explanation of each step.

### 4.1 Automatic traffic sign localization

Since fluorescent ink can only be applied to the actual surface of the traffic signs, we propose three steps to automatically identify the regions of traffic signs for precise placement of perturbations. This masking mechanism ensures that the perturbation fully complies with the shape and placement constraints of the real-world scenario. As a result, the perturbation remains effective and consistent across both digital and physical environments.

**Histogram equalization.** The histogram equalization Pizer et al. [1987] is used if the input image $x$ satisfies the following condition:

$$\frac{P_{99}(t(x(i,j))) - P_1(t(x(i,j)))}{\max(t(x(i,j))) - \min(t(x(i,j)))} < Th \tag{1}$$

where $t(x(i,j))$ represents the pixel intensity at coordinates $i$ and $j$ in the image $x$. Here, $P_{99}$ and $P_1$ are the 99th and 1st percentile of pixel values, respectively, and $Th$ is a threshold fraction.

**Canny edge detection.** We use the canny edge detector Canny [1986] with a selected threshold to detect the edge in image $x$. With customized high and low thresholds, we filter out non-edge information and highlight the edges of traffic signs in the image.

**Color-based detection.** We propose a color-based detection algorithm as follows.

We define HSV Smith [1978] ranges for each color in Table 1. Voxels falling inside these boxes are assigned a value of "255" in the output array, while those outside are assigned "0". The four color masks are merged via bitwise OR, followed by morphological opening and closing to reduce noise and fill gaps. Finally, we compare the areas identified by the two detectors. The larger area is selected as the region $A$ of the traffic sign, and the mask matrix $M_A$ is determined accordingly.

| Color | lower range | upper range |
|---|---|---|
| $Yellow$ | (20, 40, 50) | (35, 255, 210) |
| $Blue$ | (90, 40, 50) | (120, 255, 210) |
| $Red_1$ | (0, 40, 50) | (10, 255, 210) |
| $Red_2$ | (165, 40, 50) | (179, 255, 210) |
| $Black$ | (0, 40, 50) | (10, 255, 210) |

Table 1: HSV color ranges

### 4.2 Fluorescence modeling

After identifying the legal area for perturbation, we aim to simulate the effect of fluorescent ink on a traffic sign, particularly under UV light.

**Fluorescence definition.** First, we assume that the fluorescent ink is used to draw a circle, then the parameters of circle $C_0$ are defined as follows:

$$\theta_0 = ((x_0, y_0), r_0, \gamma_0, \alpha_0) \tag{2}$$

corresponding to the following aspects of the circle $C_0$:

- $(x_0, y_0) \in [W, H] \subset \mathbb{R}^2$: Coordinates of the circle in an image with width $W$ and height $H$.
- $r_0 \in [r_{min}, r_{max}] \subset \mathbb{R}$: Radius of the circle relative to the patch size.
- $\gamma_0 = [R_0, G_0, B_0] \subset \mathbb{R}^3$: RGB color triplet of the circle.
- $\alpha_0 \in [0, 1] \subset \mathbb{R}$: Opacity level of the circle.

**Perturbation function.** Let $x$ be a 2D image where $x(i, j)$ denotes the pixel at the $(i, j)$ location. We define the perturbation function for a single circle in the image, $\pi(x; \theta_0)$, as follows:

$$\pi(x; \theta_0)(i, j) = x(i, j) \cdot (1 - \alpha(i, j)) + \alpha(i, j) \cdot \gamma_0 \tag{3}$$

Intuitively, each pixel $\pi(x; \theta_0)(i, j)$ in the perturbed image is a linear combination of the original pixel and the color $\gamma_0$, weighted by the position-dependent alpha mask $\alpha_0$. To create our perturbed

image, we combine $K$ single-circle $(C_0, \cdots, C_{K-1})$ as follows:

$$\pi(x; \theta) = \pi(x; \theta_0) \circ \pi(x; \theta_1) \circ \cdots \circ \pi(x; \theta_{K-1}) \tag{4}$$

where the parameters $\theta = (\theta_0, \ldots, \theta_{K-1})$ are the concatenation of the parameters for each circle.

**Fluorescence intensity.** In this study, UV light intensity primarily affects the illumination of the traffic sign area $A$, while other components remain unchanged. Starting with a clean image $x$ in RGB color space, we first convert $x$ to LAB color space:

$$LAB(x) = [L_x, A_x, B_x] \tag{5}$$

Given masks $M_A$ and $M_F$, the value of pixel $(i, j)$ in the adversarial image $x_{adv}$ is:

$$
\begin{aligned}
LAB(x_{adv})(i,j) &= [L_{x_{adv}}^{i,j}, A_{x_{adv}}^{i,j}, B_{x_{adv}}^{i,j}] \\
&= \begin{cases}
LAB(x)(i,j) \cdot [l_1, 1, 1]^T, & (i,j) \in A \\
& \wedge (i,j) \notin F \\
LAB(x)(i,j) \cdot [l_2, 1, 1]^T, & (i,j) \in F \\
LAB(x)(i,j) \cdot [1, 1, 1]^T, & (i,j) \notin A
\end{cases}
\end{aligned}
\tag{6}
$$

Finally, we convert $x_{adv}$ back to RGB color space. We refer to the entire AE generation process as:

$$x_{adv} = LAB(\pi(x; \theta)) = LAB(\pi(x; \theta_0) \circ \cdots \circ \pi(x; \theta_K)) \tag{7}$$

### 4.3 Fluorescence optimization

In this section, we consider two customized loss functions:

$$\min_{\theta \in \Theta} \mathbb{E}_{x \sim X, t \sim T} [\ell_{goal} + \lambda \ell_{area}] \tag{8}$$

where $\theta$ is the attack parameters and $t$ denotes a random transformation. $X$ and $T$ correspond to their respective distributions, $\mathbb{E}$ denotes the expectation, and $\lambda$ is a weighting factor used to balance the different components of the loss function.

**Goal-based loss.** The goal-based loss $\ell_{goal}$ is tied to the attack objectives, allowing the attacker to apply different $\ell_{goal}$ depending on the specific attack goals.

For a hiding attack, an attacker attempts to eliminate the detection results:

$$\ell_{goal} = \Pr(object) \cdot \Pr(class) + \beta IoU_{predceted}^{truth} \tag{9}$$

where $\Pr(object)$ and $\Pr(class)$ correspond to the detector's outputs, which respectively represent: (1) the confidence that an object exists in a given cell, and (2) the classification confidence for a specific class in that cell. Additionally, the attacker minimizes the "intersection over union" (IoU) score between the predicted bounding box and the ground truth bounding box. $\beta$ is a manually set penalty term. This strategy forces the detector to inaccurately predict the bounding box location, resulting in the incorrect detection of the object's position.

For a generative attack, an attacker aims to improve the confidence by minimizing Equation 10:

$$\ell_{goal} = -\Pr(object) \cdot \Pr(class) \tag{10}$$

The attacker focuses on increasing the detector's confidence in its output and does not need to minimize the IoU score.

For a misrecognition attack, an attacker attempts to reduce the original category's score:

$$\ell_{goal} = \log(p_y) \tag{11}$$

where $p_y$ denotes the probability of the original category $y$. As $p_y$ decreases, the probabilities of other categories increase, which can lead to a change in the model's predicted category.

**Area-based loss.** To introduce the smallest perturbation possible, we minimize the loss:

$$\ell_{area} = \min_{r \in [r_{min}, r_{max}]} \sum_{i=1}^{K} \pi r_i^2 \tag{12}$$

where $K$ is the number of circles and $r$ is the radius of each circle. The goal is to make the perturbation subtle enough that the driver does not notice any anomaly, while the TSR system is led to make an incorrect decision.

**Particle swarm optimization.** In the black-box setting with discrete coordinate values in $\Theta$, we use particle swarm optimization (PSO) Kennedy and Eberhart [1995]. PSO is robust to the initial settings, aligning with our use of random initialization for parameters like perturbation position and color. To enhance success rates, we employ the $n$-random-restarts strategy, allowing us to reinitialize and rerun the PSO up to $n-1$ times if the attack fails.

### 4.4 Robustness improvement

In this section, we propose two methods to improve the robustness of FIPatch in the real world.

**Expectation over transformation.** EOT Athalye et al. [2018] is an effective method for addressing discrepancies between digital and real-world scenarios. In this paper, we extend EOT's transformation distributions $\mathcal{T}$ to accommodate variations in fluorescent materials across different physical environments. This includes accounting for perspective, brightness, and other environmental factors previously overlooked, as detailed in Appendix B.

**Transparency transformation.** To simulate the transparency of fluorescent ink when not triggered, we apply a specific alpha value. Over time, environmental factors reduce the ink's transparency, impacting stealthiness. We model this over 5 days, with transparency ranging from [0, 0.1]. Despite this, the effect remains invisible to passing vehicles in Appendix Figure 12.

## 5 Evaluation

In this section, we evaluate the attack's performance in the physical world and provide the results in the digital world in Appendix C.

### 5.1 Experimental setup

**Datasets.** We select two datasets of traffic signs captured in real driving conditions: the German Traffic Sign Recognition Benchmark (GTSRB) Stallkamp et al. [2012] and the Chinese Traffic Sign Recognition Database (CTSRD) Huang. These datasets are widely used in current research Liu et al. [2019] Li et al. [2020] Ye et al. [2021].

**Models.** We evaluate 10 different models. For traffic sign detection, we use Yolov3 and Faster R-CNN, both pre-trained on the COCO dataset Microsoft [2018]. We set the input size for the traffic sign detection models to $416 \times 416$ and the confidence threshold for the output boxes to $0.5$. For traffic sign classification, we train CNN, Inception v3, MobileNet v2, and GoogleNet on the GTSRB dataset. Additionally, we use ResNet50, ResNet101, VGG13, and VGG16 for the CTSRD dataset. We set the input size to $32 \times 32$ for all classifiers, except Inception v3, which uses an input size of $299 \times 299$. The number of queries required to perform the attack is 1500.

**Metrics** We define the attack success rate (ASR) Equation 13 to evaluate FIPatch attacks.

$$ASR = \frac{1}{N} \sum_{1}^{N} I_{F(x,untri)=y \& F(x,tri)\neq y}(x) \quad (13)$$

where $N$ is the total number of frames or input images, $I$ is the indicator, $F$ represents the model's prediction function, and $y$ is the original prediction label. The indicator $I(x)$ equals 1 if the model's prediction is $y$ when the attack is not triggered, and 0 if the model misclassifies when the attack is triggered.

**Devices.** As shown in Figure 3, we employ a Tesla Model Y as the victim's vehicle, equipped with a dashcam recording videos at 30 fps.

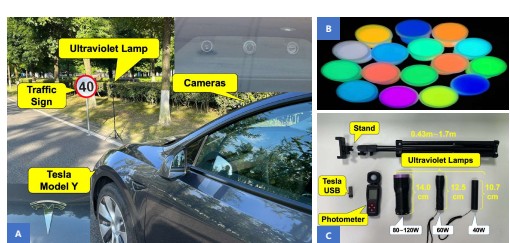

Figure 3: Experimental setup in real-world environments. (A) A traffic sign in front of a Tesla Model Y with a UV lamp. The Tesla's front cameras capture videos sent to the TSR system. (B) Sixteen fluorescent material colors. (C) Equipment: stand, Tesla USB, photometer, and 3 UV lamps.

Table 2: The ASR of FIPatch on various models in the physical world.

| Ambient Light (Lux) | Frames | Generative Attack | | Hiding Attack | | Misrecognition Attack | | | |
|---|---|---|---|---|---|---|---|---|---|
| | | Yolov3 | Faster R-CNN | Yolov3 | Faster R-CNN | ResNet50 | VGG13 | MobileNet v2 | GoogleNet |
| 200 | 4374 | 98.31% | 98.66% | 91.59% | 87.81% | 100% | 99.82% | 98.93% | 100% |
| 500 | 3655 | 98.72% | 93.41% | 83.64% | 80.09% | 99.35% | 98.01% | 95.38% | 97.52% |
| 1000 | 4163 | 95.22% | 88.31% | 69.19% | 64.91% | 94.26% | 92.58% | 92.15% | 93.66% |
| 2000 | 3719 | 94.06% | 86.10% | 53.81% | 48.43% | 90.59% | 86.37% | 85.92% | 84.03% |
| 3000 | 3924 | 89.63% | 83.39% | 31.48% | 25.92% | 84.12% | 79.55% | 74.59% | 76.15% |

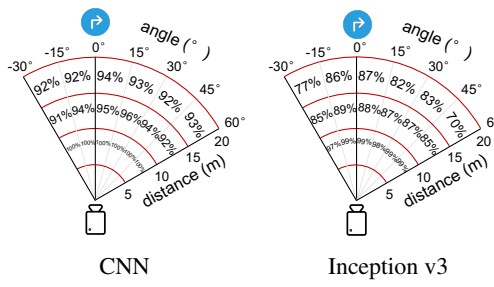

| CNN | Inception v3 | CTSRD | GTSRB |

Figure 5: Impact of the distance & angle between vehicle and traffic sign on ASRs.

Figure 6: Impact of the UV lamp power on ASRs.

These videos are transmitted to a computer to simulate the TSR system. Fluorescent ink is applied to a traffic sign positioned 1.5 meters high, and a UV lamp is mounted on a stand 2.0 meters away. All models are deployed on an Apple M2 Pro GPU.

## 5.2 Overall performance

In this section, we present examples and experimental results of FIPatch in the physical world. In Figure 4, examples of three different attacks are illustrated. Specifically, when the attack is untriggered, the model performs detection and recognition normally. For example, it recognizes a blank sign as empty; it correctly recognizes various traffic signs such as stop signs and turn right ahead. When an attacker triggers an attack, the generative attack allows the model to incorrectly detect the stop sign by drawing a simple border. The hiding attack makes the model fail to detect the sign without affecting the naked eye's recognition. The misrecognition attack induces the model to misclassify the traffic signs. We emphasize that in the physical world, the perturbation only needs to be active for a short time. When a self-driving car approaches the stop sign, even if it fails to recognize the stop sign for merely a short time window, it can lead to a fatal accident.

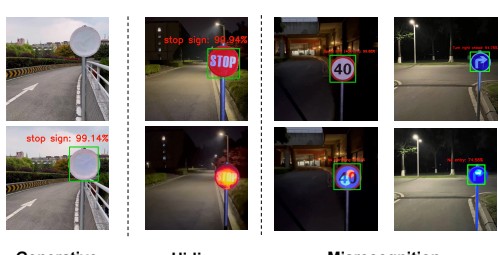

**Generative** **Hiding** **Misrecognition**

Figure 4: Untriggered (top) and triggered (bottom) attack examples in real-world environments.

We measure the ambient light intensity on traffic signs using a photometer and analyze the ASRs of FIPatch under various light conditions. As shown in Table 2, Yolov3 exhibits higher ASR compared to Faster R-CNN for both generative and hiding attacks, suggesting that Faster R-CNN is more robust against FIPatch. Generative attacks are more effective than hiding attacks at all ambient light levels while hiding attacks maintain ASRs above 80% only when the light is below 500 lux. At 3000 lux, the ASR for hiding attacks drops to below 32%. This drop is likely because detectors are more sensitive to perturbations on blank signs, such as contours, but more resilient when predicting existing traffic signs. All four classification models are highly vulnerable to attacks, with ASRs above 93% when light is below 1000 lux. Overall, the ASRs decreases with increasing ambient light because the fluorescent effect diminishes, reducing its impact on the models.

### 5.2.1 Impact of FIPatch in real-world enviroments

**Impact of distance & angle.** We place traffic signs at various locations and record around 3000 frames of video with a dashcam. The setup includes a traffic sign of size $60\,\mathrm{cm} \times 60\,\mathrm{cm}$, ambient brightness of approximately 200 lux, and a fluorescent perturbation radius of $14\,\mathrm{cm}$. The distance between the camera and the traffic sign ranges from $0\,\mathrm{m}$ to $20\,\mathrm{m}$, and angles follow the perspective described in Appendix B. As shown in Figure 5, CNN models achieve an ASR of over $91\%$ across all distances, although ASR decreases with distance and is less affected by angle variations. In contrast, Inception v3 experiences a significant drop in ASR, achieving only $70\%$ or $77\%$ at the furthest distances, indicating higher sensitivity to angle changes at greater distances. Overall, while increasing distance reduces ASR for all models, angle variations have less impact except at extreme angles ($60°$ or $-30°$). Additional results on other models can be found in Appendix Figure 16.

**Impact of UV lamp power.** To examine the effect of UV lamp power on ASRs, we test three types of UV lamps: 40 Watts (W), 60W, and three levels of 80W, 100W, and 120W. As shown in Figure 6, VGG13 shows the greatest sensitivity to changes in UV lamp power. When using a 40W UV lamp, Inception v3 achieves the lowest ASR of $77\%$. In contrast, with a 120W UV lamp, all models achieve ASR above $97\%$. Overall, the ASR increases with UV lamp power because higher power results in a brighter fluorescent effect, leading to stronger interference with the models.

**Impact of vehicle speed.** For detection, we perform generative attacks on Yolov3 and Faster R-CNN. For classification, we apply misrecognition attacks to four classification models. As shown in Figure 3, Yolov3 consistently shows higher ASRs than Faster R-CNN across all vehicle speeds. At a speed of 15 km/h, Faster R-CNN achieves only a $77\%$ ASR. For classi-

Table 3: Impact of the vehicle speed on ASRs.

| Speed | Model | | | | | |
|---|---|---|---|---|---|---|
| (km/h) | Yolov3 | Faster R-CNN | ResNet50 | VGG13 | CNN | GoogleNet |
| 0 | 98% | 91% | 99% | 98% | 100% | 100% |
| 5 | 96% | 90% | 98% | 96% | 99% | 93% |
| 10 | 91% | 85% | 96% | 95% | 97% | 87% |
| 15 | 83% | 77% | 93% | 87% | 97% | 83% |

fiers, the ASRs for all four models remains above $93\%$ when the vehicle speed is below 10 km/h. Notably, ResNet50 and CNN maintain stable ASRs, while other models experience a significant drop in ASRs when the speed exceeds 10 km/h. This decline is attributed to rapid changes in the angle of the traffic sign and reflections from the fluorescent material, which make it more challenging for the models to maintain accurate detection as speed increases.

We also evaluate the impact of distance between UV lamp and traffic sign in Appendix D and find that there is an ASR of at least $81\%$ at a distance of 8 meters. This distance is sufficient for an attacker to place the UV lamp in the bushes near the traffic sign. In addition, since the UV light is invisible to the naked eye, the victim cannot locate the light source, and the attack device is in the bushes, further increasing the stealthiness of the attack. For elevated traffic signs commonly found on highways, attackers can leverage drones to carry out the attack. By equipping a drone with a UV lamp and hovering it at a specific position, the attacker can activate the perturbation without physical access to the sign. The drone only needs to remain in position for a few seconds during the attack and can quickly leave the area afterward, making the attack highly stealthy and flexible. Other factors, e.g., radius, colors, position, and shape, are evaluated in Appendix C.1.1.

## 6 Defenses

Since defenses against object detectors are not well-explored, we focus on misrecognition attacks, as detailed in Table 4. The image smoothing Cohen et al. [2019] shows minimal impact on the ASRs, with a slight increase observed for some models. The feature compression Jia et al. [2019] offers slight benefits for ResNet50, CNN, and GoogleNet, reducing the ASRs by $1 \sim 2\%$. The input randomization Xie et al. [2017] does not effectively mitigate fluorescent perturbations, thus having little impact across all models. The adversarial training Madry et al. [2017], known to be effective in other scenarios, does not significantly defend against our scheme. This is because attackers can vary colors and perturbation positions, preventing models from effectively learning our attack patterns. The defensive dropout Wang et al. [2018] enhances model robustness by reducing network complexity. As seen in Table 4, this method provides the most effective defense against our attacks compared to other techniques. In summary, current popular defense methods are ineffective against FIPatch attacks, presenting new challenges to driving safety and security. For more details, refer to Appendix E.

Table 4: The ASR of FIPatch across various defenses.

| Model | W/o defense | Image Smoothing | Feature Compression | Input Randomization | Adversarial Training | Defensive Dropout |
|-------|-------------|-----------------|---------------------|---------------------|----------------------|-------------------|
| ResNet50 | 99.47% | 98.54%(-0.93%) | 97.82%(-1.65%) | 99.18%(-0.29%) | 98.63%(-0.84%) | 97.17%(-2.30%) |
| ResNet101 | 99.30% | 99.46%(+0.16%) | 98.47%(-0.83%) | 98.35%(-0.95%) | 99.27%(-0.03%) | 97.55%(-1.75%) |
| VGG13 | 99.29% | 99.11%(-0.18%) | 99.04%(-0.25%) | 98.60%(-0.69%) | 98.75%(-0.54%) | 97.92%(-1.37%) |
| VGG16 | 99.81% | 98.74%(-1.07%) | 99.22%(-0.59%) | 98.92%(0.90%) | 99.04%(-0.77%) | 98.33%(-1.48%) |
| CNN | 100% | 99.87%(-0.13%) | 98.61%(-1.39%) | 99.28%(-0.72%) | 99.84%(-0.16%) | 99.34%(-0.66%) |
| Inception v3 | 98.75% | 99.14%(+0.39%) | 98.36%(-0.39%) | 98.54%(-0.21%) | 99.17%(+0.42%) | 96.72%(-2.03%) |
| MobileNet v2 | 99.32% | 98.72%(-0.60%%) | 98.45%(-0.87%) | 99.04%(-0.28%) | 98.66%(-0.66%) | 98.17%(-1.15%) |
| GoogleNet | 99.68% | 99.54%(-0.14%) | 97.55%(-2.13%) | 98.26%(-1.42%) | 98.52%(-1.16%) | 97.24%(-2.44%) |

Due to the vulnerability of the TSR system, we propose two potential defenses against our FIPatch attack. One defense strategy is using high-definition maps, which provide stable, accurate traffic sign information unaffected by FIPatch. These maps, regularly updated by providers based on official traffic authority announcements, allow vehicles to make informed decisions even if signs are altered. However, they don't replace the need for sensors in case of unexpected conditions or map delays. Another promising defense is collaborative perception among multiple vehicles. If one vehicle is attacked by fluorescent ink, nearby vehicles can share their recognition results, enabling the affected vehicle to correct its prediction, enhancing overall system robustness.

## 7 Discussions

**Limitations.** Our FIPatch has two limitations. First, our outdoor experiments mainly assess the effects at the AI component level rather than the autonomous vehicle system level. Second, ambient light impacts different attack goals in varying ways.

**Future work.** In our future work, we plan to focus on two main directions. First, we intend to investigate the use of fluorescent materials to challenge object detection systems. Specifically, adding suitable perturbations using fluorescent materials on curved surfaces presents a significant challenge. Second, we aim to develop effective defenses against FIPatch. This includes exploring multi-vehicle collaboration and leveraging deep learning models to enhance security. Determining how to implement these defenses effectively remains a challenge and will be a key focus of our future research.

**Societal impacts.** The proposed FIPatch attack has important societal implications. On the positive side, it exposes a previously overlooked vulnerability in TSR systems, thereby encouraging the development of more robust models and defenses. However, due to its stealthiness and physical feasibility, FIPatch also poses potential risks if maliciously exploited to disrupt traffic environments or compromise public safety. We emphasize that our study serves as an effective approach to identifying security issues, encouraging researchers to focus more on the robustness of models.

**Ethics statements.** The closed roads used for physical world experiments have obtained IRB approval from our institution for data collection. All images and videos used in physical world attacks are legally obtained from vehicle owners and do not contain any personal information. We ensure that there are no pedestrians during the experiments while the vehicle is operated by the driver without any safety incidents.

## 8 Conclusion

In this paper, we propose FIPatch, a stealthy and effective adversarial attack that leverages fluorescent ink to create adversarial examples in the physical world. We focus on the context of traffic sign recognition, where the goal of the attack is to alter the appearance of a traffic sign using specially crafted fluorescent ink, causing the traffic sign recognition system to either fail to detect or misclassify the sign.

Considering the physical constraints of applying fluorescent ink to multiple traffic signs under various conditions, we develop a tailored approach to create robust black-box adversarial examples. We evaluate our proposed attack method against 10 state-of-the-art detectors and classifiers in both the digital and physical worlds. Our investigation into various factors affecting the success rate of FIPatch attacks demonstrates its robustness in real-world scenarios. Finally, our analysis of existing defenses shows that current methods against adversarial examples are ineffective against FIPatch, highlighting the need for further research into this potent new attack vector.

# 9  Acknowledgements

This work is supported by the National Key R&D Program of China under Grant 2022YFB3103500, the National Natural Science Foundation of China under Grant 62020106013, the National Natural Science Foundation of China under Grant 62502075, the Sichuan Science and Technology Program under Grant 2024ZHCG0188, the Chengdu Science and Technology Program under Grant 2023-XT00-00002-GX.

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

# A   Feasibility Study

In this section, we explore the feasibility of using fluorescent ink to attack a TSR system. We introduce the fundamental concepts of fluorescent materials. Following this, we render the fluorescent effect and apply it to traffic signs, which are then analyzed by a TSR system.

## A.1   Fluorescent materials

Fluorescence occurs in certain molecules, called fluorophores (typically polyaromatic hydrocarbons or heterocycles), through a three-stage process illustrated in the jablonski energy diagram Jablonski [1933] (Figure 7).

First (①), a photon with energy $hv_{EX}$ from an external source with wavelength $\lambda_{EX}$ (like an incandescent lamp or laser) is absorbed by the fluorophore, creating an excited singlet state $S_1$.

Second (②), during its excited state, which lasts a few nanoseconds, the fluorophore undergoes conformational changes and interacts with its environment. These interactions cause energy dissipation, resulting in a relaxed singlet state $S_2$, from which fluorescence emission occurs. Not all molecules return to the ground state $S_0$ via fluorescence. Some molecules are depopulated through processes like collisional quenching and intersystem crossing.

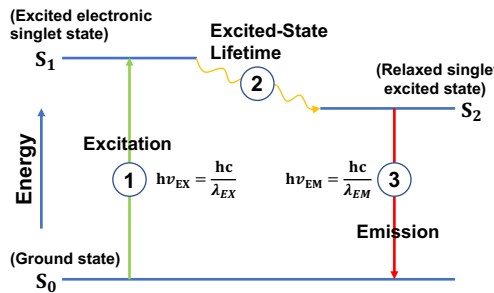

Figure 7: The jablonski energy diagram Jablonski [1933] illustrating the fluorescence process.

Finally (③), a photon with energy $hv_{EM}$ is emitted, returning the fluorophore to its ground state $S_0$. Due to energy dissipation, the emitted photon has lower energy and a longer wavelength than the excitation photon $hv_{Ex}$. The difference, $(hv_{Ex} - hv_{EM})$, is called the stokes shift Stokes [1852].

Fluorescent materials can be either solid or liquid. Solid materials like phosphors are difficult to attach to targets and lack stealthiness. This paper focuses on fluorescent ink, which is transparent when not triggered, hard to detect, and easy to apply to targets.

## A.2   Fluorescent materials rendering

In this section, we introduce the main parameters of fluorescent materials and how they render fluorescent effects on the surfaces of objects. The key parameters are fluorescence quantum yield, fluorescence excitation spectrum, and fluorescence emission spectrum. The fluorescence quantum

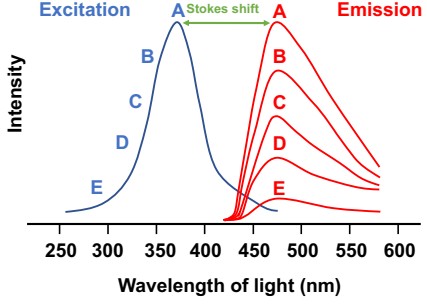

Figure 8: The effect of excitation (blue) at different wavelengths on the fluorophore emission (red) at various excitation wavelengths is as follows: (A) Excitation at the fluorophore's excitation maximum yields maximum emission. (B-E) Excitation at suboptimal wavelengths leads to decreased emission intensity, proportional to the reduced amount of excitation input.

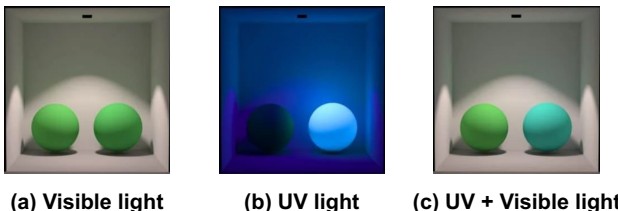

**(a) Visible light**    **(b) UV light**    **(c) UV + Visible light**

Figure 9: Examples of simulated renderings include (a) BSDF rendering in visible light, (b) fluorescent BSDF rendering in UV light, and (c) fluorescent BSDF rendering in UV and visible light.

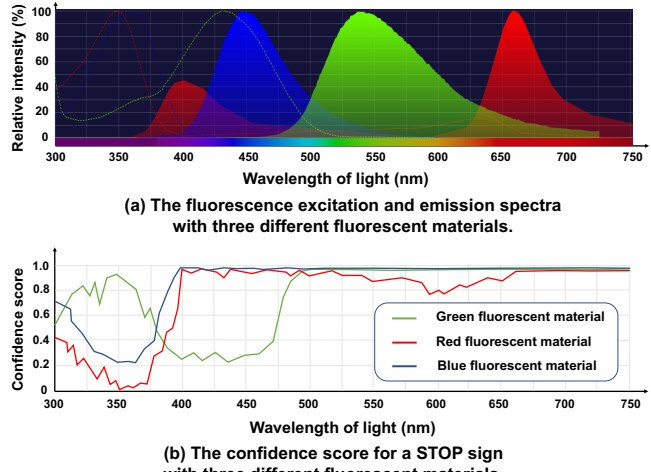

**(a) The fluorescence excitation and emission spectra with three different fluorescent materials.**

**(b) The confidence score for a STOP sign with three different fluorescent materials.**

Figure 10: The feasibility experiments involve (a) obtaining fluorescence spectra of fluorescent materials at various wavelengths, and (b) assessing the confidence scores of the TSR system regarding the STOP sign at different wavelengths for three fluorescent materials applied on its surface.

yield is the ratio of the number of fluorescence photons emitted to the number of photons absorbed. It measures the efficiency of the fluorescence process. A fluorescence excitation spectrum is obtained by fixing the emission wavelength (typically at the maximum emission intensity) and scanning the excitation wavelength. Since excitation leads to the molecule reaching the excited state upon absorption, the excitation spectrum effectively represents the absorption characteristics. A fluorescence emission spectrum is obtained by fixing the excitation wavelength and scanning the emission wavelength to produce a plot of intensity versus emission wavelength. For instance, if we fix the excitation at wavelength B (350 nm) in Figure 8 and scan the emission spectrum between 430 nm and 580 nm, we obtain the emission spectrum corresponding to wavelength B. It is important to note that illuminating a fluorophore at its excitation maximum produces the greatest fluorescence output. However, illuminating at other wavelengths only affects the intensity of the emitted light, without changing the range or overall shape of the emission profile.

To render fluorescent effects on the surface of objects, we use the bidirectional scattering distribution function (BSDF) Bartell et al. [1981], a general representation of the optical properties of surface reflection and transmission. Utilizing the *Ocean* light simulator Digital, we incorporate the specific parameters mentioned above into the fluorescence BSDF model. Figure 9 shows multiple examples of rendering. The left ball is a control sample without fluorescence, while the right ball represents a fluorophore. Note that this fluorophore is hypothetical and created solely for demonstration purposes.

### A.3   TSR with fluorescent materials

To investigate the feasibility of fooling a TSR system, we separately render three common colors, i.e. red, green, and blue, of fluorescent materials onto the surface of a traffic sign and feed the resulting images into the TSR system. The excitation and emission spectra of the fluorescent materials are

depicted in Figure 10 (a). The optimal trigger wavelengths for red, blue, and green fluorescent materials are 348 nm, 360 nm, and 430 nm, respectively.

In this section, we use standard stop signs and manually annotate their locations in images. We then render different fluorescent materials to these signs in the digital world and submit the modified images to the Yolov3 model for recognition. The model's confidence scores for detecting stop signs are shown in Figure 10 (b). The results show that fluorescent materials can effectively lower the model's confidence in recognizing stop signs, confirming the feasibility of this attack. Red fluorescent material significantly reduces the confidence score more than green and blue, likely due to the model's sensitivity to longer wavelengths. Green fluorescent material also lowers the confidence score over a wider wavelength range, thanks to its broad excitation spectrum.

From the above experiments, we can draw the following conclusions: First, a TSR system can be successfully attacked using fluorescent materials. However, the success of such an attack is not guaranteed, as the confidence scores are highly sensitive to the wavelength used. Second, various factors—such as fluorescence intensity, perturbation placement, and ambient light—significantly affect the attack's effectiveness. Therefore, the same set of fluorescence parameters cannot be universally applied to different traffic signs.

# B  Different Settings for EOT

We present the transformation methods used in EOT.

**(1) Background.** We select various backgrounds, including highways, viaducts, city lanes, and country roads, and carefully position traffic signs at the road edges. Following the approach of using Google Images as suggested in Zhao et al. [2019], we gather a diverse set of road backgrounds to further expand the transformation set.

**(2) Brightness.** To simulate varying ambient brightness, we capture images of traffic signs under different weather conditions and times of day, including sunny, cloudy, rainy, evening, night, and dawn. Additionally, the intensity of headlights affects the visibility of traffic signs. We convert the traffic sign image to LAB color space, adjust the L channel values within the range [0, 50], and then convert the image back to RGB.

**(3) Perspective.** As shown in Figure 11, we apply perspective transformations based on real-world environments in two ways. First, traffic signs are typically positioned on the right side or above the road (in right-driving countries), so they are rarely seen on the right side of the screen. Therefore, we set the horizontal field of view to range from 30° left to 60° right, with 0° directly in front.

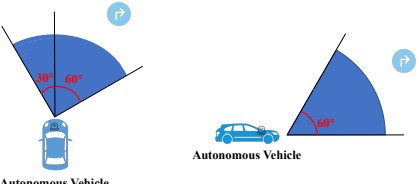

Figure 11: Horizontal (left) and vertical (right) viewing angles.

Second, On-board cameras are usually installed at specific heights (e.g., the forward-looking camera on a Tesla Model Y is positioned 1.4 to 1.5 meters above the ground). Traffic sign heights also have specific requirements (e.g., 4 to 17 feet in the U.S. on Uniform Traffic Control Devices [2023.12]), typically aligning with the camera height. Thus, we set the vertical field of view to range from 0° to 60°.

**(4) Distance.** As an autonomous vehicle approaches a traffic sign, the size of the sign in the camera's view increases progressively. We account for the size of traffic signs at varying distances during the optimization process. Specifically, we set the maximum distance between the vehicle and the traffic sign to 20 meters and record the sign's size at different distances.

**(5) Rotation.** Traffic signs are not always directly in front of the vehicle's cameras and may be slightly offset. These slight rotations can cause misrecognition by the model, so we account for rotations of plus or minus 10 degrees during the optimization process.

**(6) Motion.** During vehicle travel, images captured by the camera may suffer from motion blur caused by road bumps and other factors. To enhance the model's robustness, we simulate motion blur at various angles and directions.

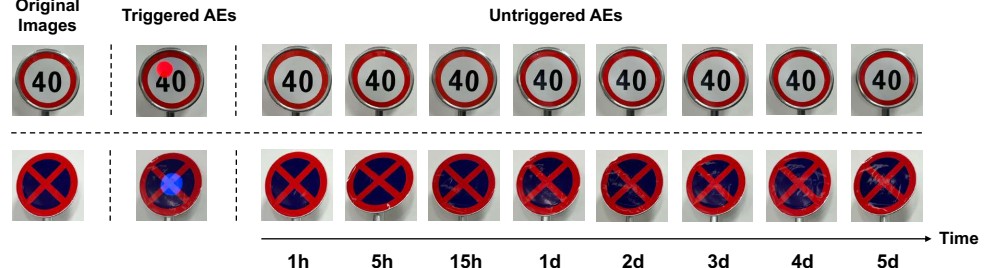

Figure 12: Transparency of fluorescent ink at different placement times when no attack is triggered.

Table 5: The ASR of FIPatch on various models in simulation.

| Overall Performance | | | | | | | | | | |
|---|---|---|---|---|---|---|---|---|---|---|
| & Transferability | | ResNet50 | ResNet101 | VGG13 | VGG16 | | | CNN | Inception v3 | MobileNet v2 | GoogleNet |
| Source Model | ResNet50 | *100%* | 95% | 89% | 91% | CNN | *100%* | 76% | 79% | 80% |
| | ResNet101 | 93% | *100%* | 92% | 94% | Inception v3 | 92% | *100%* | 69% | 81% |
| | VGG13 | 94% | 85% | *99%* | 92% | MobileNet v2 | 92% | 83% | *99%* | 87% |
| | VGG16 | 95% | 93% | 88% | *100%* | GoogleNet | 91% | 81% | 80% | *100%* |
| Original Accuracy | | 99.27% | 99.11% | 98.70% | 99.19% | | 99.27% | 99.33% | 99.49% | 99.61% |

# C  Digital-world attacks

## C.1  Overall performance

We conduct misrecognition attacks on various classifiers in the digital domain. Specifically, we configure the PSO search color space from $(0,0,0)$ to $(255,255,255)$. The setup includes a single circle with a radius ranging from 0 to 15 and a fluorescent effect transparency set between 0.7 and 0.9. We perform 5 random restarts and run 30 iterations per PSO.

We present the experimental results in Table 5 in two parts. First, the underlined results show the ASRs of our method in a black-box setting. By using optimal attack parameters for each traffic sign, we achieve nearly $100\%$ ASRs on several high-precision models, with VGG13 and MobileNet v2 having slightly lower ASRs of $99\%$. Second, we evaluate the transferability of our attack. In this context, the source model is the attacker's shadow model, and the target model is the one intended for attack. As shown in Table 5, the ASRs are above $85\%$ for ResNet50, ResNet101, VGG13, and VGG16. The lowest ASR of $69\%$ is observed when transferring from Inception v3 to MobileNet v2. Models with similar architectures, such as ResNet50 to ResNet101, achieve high ASRs, reaching up to $95\%$. These results demonstrate the effectiveness of our attack across various models.

### C.1.1  Impact of FIPatch in simulation

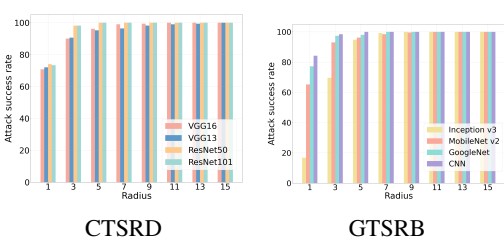

CTSRD                GTSRB

Figure 13: Impact of the radius on ASRs for models trained on CTSRD and GTSRB.

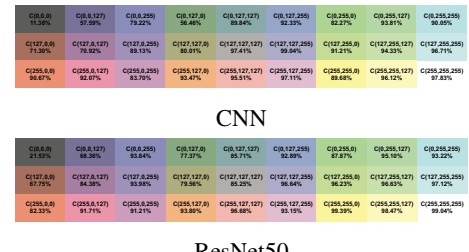

Figure 14: Impact of the colors $C(r, g, b)$ on ASRs.

**Impact of radius.** To investigate the effect of the fluorescent ink radius, we vary it from 1 to 15 pixels relative to the image height. As shown in Figure 13, there is a strong correlation between the perturbation radius and the ASR: a larger radius generally leads to a higher ASR. Additionally, smaller radii have less impact on more complex models.

Table 6: Impact of the number of circles on ASRs.

| The number | Model | | | | | | | |
|---|---|---|---|---|---|---|---|---|
| of circle | ResNet50 | ResNet101 | VGG13 | VGG16 | CNN | Inception v3 | MobileNet v2 | GoogleNet |
| 1 | 88.42% | 89.74% | 84.18% | 84.31% | 96.11% | 49.26% | 95.60% | 98.32% |
| 2 | 95.24% | 97.86% | 94.79% | 93.82% | 94.19% | 75.57% | 89.10% | 95.28% |
| 3 | 96.41% | 100% | 97.36% | 95.43% | 98.23% | 76.91% | 92.15% | 96.28% |
| 4 | 96.41% | 100% | 96.75% | 95.38% | 97.83% | 80.17% | 97.88% | 96.42% |
| 5 | 98.27% | 100% | 98.65% | 97.53% | 97.14% | 78.26% | 95.70% | 97.49% |

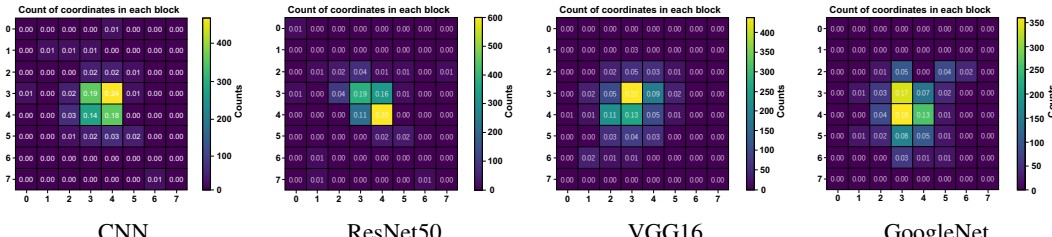

Figure 15: Impact of the positions on ASRs.

**Impact of colors.** To analyze the impact of color on the attack's effectiveness, we test 27 different colors. As shown in Figure 14, black color results in the lowest ASR for both models and fluorescent materials that emit black are not found in reality. For the CNN model, the color $C(127, 127, 255)$ yields the highest ASR, while for ResNet50, the color $C(255, 255, 0)$ achieves the highest ASR.

**Impact of number of circles.** We next examine how the number of circles affects the ASR. As shown in Table 6, while adding more circles generally improves the ASRs, the effect varies across different models. This is because increasing the number of circles does not necessarily enlarge the perturbation area, and some circles may overlap.

**Impact of positions.** To explore how perturbation positions affect the ASRs, we divide the $32 \times 32$ pixel image into 64 blocks of $8 \times 8$ pixels each. As shown in Figure 15, the central region of the image has the highest percentage of successful attacks. This is because the traffic signs in the dataset are centered, making attacks on the edges less effective. Consequently, the center of the sign is the most vulnerable and prone to successful attacks.

**Impact of shapes.** In this section, we analyze the impact of different perturbation shapes on the ASRs. As shown in Figure 17, circles achieve a much higher ASRs compared to straight and curved lines. This is because circles cover a larger area and have a more significant impact on misclassifying models. In contrast, straight lines and curves result in ASRs below $60\%$ on Inception v3, indicating that these simple linear perturbations are less effective in causing misclassification.

## D Impact of distance between UV lamp and traffic sign

We place traffic signs at $30°$ to the right of the vehicle at a distance of $10\,\mathrm{m}$. The ambient light intensity is 150 lux, the radius of the patch remains $14\,\mathrm{cm}$ and the UV lamp is positioned directly in front of the traffic sign. We adjust the distance between UV lamp and traffic sign to test its impact on the ASR of each model. As shown in Table 7, all four models achieve a 100% ASR when the distance is less than $1\,\mathrm{m}$. As the distance increases, the ASR gradually decreases, as the fluorescence effect triggered by the UV light weakens with greater distance. ResNet50, VGG13, and CNN show

Table 7: Impact of the distance between UV lamp and traffic sign on ASRs.

| Model | 1m | 2m | 4m | 6m | 8m |
|---|---|---|---|---|---|
| ResNet50 | 100% | 98% | 96% | 90% | 87% |
| VGG13 | 100% | 97% | 96% | 89% | 84% |
| CNN | 100% | 100% | 98% | 93% | 90% |
| GoogleNet | 100% | 93% | 89% | 87% | 81% |

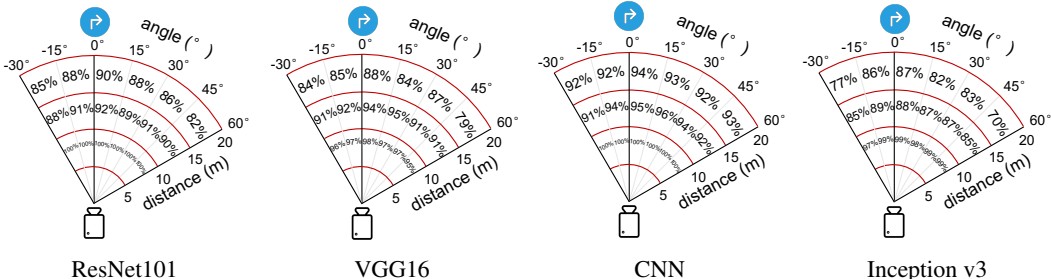

| ResNet101 | VGG16 | CNN | Inception v3 |
|:---:|:---:|:---:|:---:|

Figure 16: Impact of the distance & angle between vehicle and traffic sign on ASRs.

the most significant decrease in ASR when the distance increases from $4\,\mathrm{m}$ to $6\,\mathrm{m}$. In contrast, GoogleNet experiences the largest drop in ASR between $2\,\mathrm{m}$ and $4\,\mathrm{m}$. To achieve optimal attack results, we recommend placing the UV lamps in bushes within 4m of the traffic signs. Notably, all models maintain an ASR of at least 81% even at a distance of $8\,\mathrm{m}$.

## E  Defenses

Typically, AE defenses are designed for digital domains to detect small perturbations. In physical AEs, attackers cannot precisely control inputs and are constrained by real-world conditions. Since defenses for physical AEs are less well-studied compared to those for digital AEs, and many existing approaches simply apply general AE defenses to the physical world, we select three popular AE defense classes to evaluate our FIPatch.

The first category is input preprocessing, which includes image smoothing Cohen et al. [2019], feature compression Jia et al. [2019], and input randomization Xie et al. [2017]. Specifically, image smoothing Cohen et al. [2019] involves training a neural network $f$ with Gaussian data augmentation (variance $\sigma^2$) and using $f$ to create a new "smoothing classifier." In this paper, we set $\sigma$ to 0.5. Feature compression Jia et al.

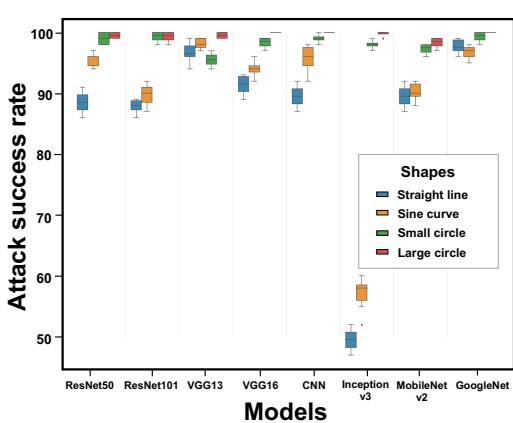

Figure 17: Impact of the shapes on ASRs.

[2019] leverages redundant information in images to defend against AEs. We use the same network structure as in Jia et al. [2019] and apply it to AEs generated by our FIPatch. Input randomization Xie et al. [2017] uses random resizing or padding to reduce adversarial effects. We resize the input image from $32 \times 32$ to $36 \times 36$. For Inception v3, we first shrink the image to $290 \times 290$ and then pad it to $299 \times 299$.

The second category is adversarial training Madry et al. [2017], a widely used defense method that aims to help the model learn to recognize and counteract attacks. We set the perturbation radius in our FIPatch to 7, with Particle Swarm Optimization (PSO) exploring various locations and colors. Each model generates AEs that successfully perform an attack, representing $10\%$ of the initial training set, and records the original correct labels of these AEs. Each model then continues training for 10 epochs on its own set of generated AEs.

The third category is structural modifications, with defensive dropout Wang et al. [2018] being a notable example. This method improves upon random activation pruning. We implement dropout during both training and testing, setting the dropout rate to 0.3 to achieve robust defense.

Due to the vulnerability of the TSR system, we propose a potential defense against our FIPatch attack: high definition maps. Traffic signs are typically fixed, unlike changeable traffic lights, which means their information remains stable over time. High definition maps, which contain accurate

traffic sign information, are not affected by FIPatch attacks. Vehicles equipped with such maps can make informed decisions based on the traffic sign data provided, independent of potential perturbations caused by attacks. Furthermore, changes to traffic signs generally require approval from the traffic department. High definition map providers can update the map information in real-time based on official announcements from traffic authorities, ensuring that the data remains current and reliable. However, using high definition maps does not eliminate the need for a sensing system. In cases of unexpected road conditions or delays in map updates, sensors are crucial for ensuring safe driving. Therefore, combining high-definition maps with robust sensor systems improves the safety of autonomous vehicles against AEs.

