# OpenReview forum: "The Fluorescent Veil: A Stealthy and Effective Physical Adversarial Patch Against Traffic Sign Recognition"
_NeurIPS.cc/2025/Conference — NeurIPS 2025 poster_

### Official Review · Reviewer_7D6S · 2025-06-23

**Clarity:** 4
**Significance:** 4
**Originality:** 3
**Rating:** 6
**Confidence:** 5

**Summary:**

This paper proposes a novel adversarial attack method for TSR systems using fluorescent ink as the attack medium. The perturbation remains invisible under normal lighting conditions and only becomes active under invisible UV light. Both digital and physical experiments are conducted, including tests on real vehicles, to verify the effectiveness and robustness of FIPatch. Additionally, the paper evaluates the proposed method against existing defense mechanisms.

**Questions:**

This work discovers a novel attack medium (fluorescent ink) to misclassify TSR while being unnoticed. The problem is not new, and previous work has shown how to perform physical attacks. Nonetheless, the paper discovers how fluorescent ink patches can be designed to further advance the field. Experiments in the physical world confirm the effectiveness of the attack. Also, the use of real vehicle cameras and machine models enhances the credibility and practicality of the attack.

The writing of the paper is logically clear and easy to follow. I really appreciated the explanation of fluorescence, even though it is placed in the appendix. I also found the simulation results, especially the ablation studies, very helpful for understanding the experimental setup. The authors might consider moving part of this content into the main body of the paper. Still, I have concerns about FIPatch.

UV light is invisible, using a UV lamp may be visible. How can the attacker ensure that the UV lamp remains covert? Moreover, in some scenarios, such as highways, where traffic signs are usually placed high, can the attack still be successful at this point? It would be helpful if this aspect could be explained to alleviate the concerns about the stealthiness.

The paper should further discuss the feasibility of defenses, e.g., could a UV filter defend against the attacks in this paper? Relying solely on high-definition maps may not be sufficient, especially in cases where traffic signs have been replaced by workers but the maps are not yet updated.

Although physical experiments are thorough, the impact is only measured at the AI component level (classification/detection), not on full AV systems (e.g., driving decisions).

**Ethical Concerns:**

["NO or VERY MINOR ethics concerns only"]

**Final Justification:**

Thanks for your response. I have read the authors’ rebuttal and the comments from other reviewers. I appreciate the additional experimental results and the discussion addressing my concerns so I raised my score.

**Limitations:**

Yes, the authors have explicitly discussed limitations and societal impacts in Section 7. They acknowledge that (1) the attack was not evaluated in full driving systems, and (2) ambient light significantly affects performance. The ethical section appropriately addresses safety and IRB compliance, showing responsible disclosure.

**Paper Formatting Concerns:**

None.

**Quality:**

3

**Strengths And Weaknesses:**

Strengths:
1. The use of fluorescent materials as an adversarial medium is both novel and practical.
2. The methodology is well-motivated and technically sound.
3. The experiments include both digital and physical results. The use of real vehicles enhances the credibility of the method's applicability and effectiveness in the physical world.

Weaknesses:
1. The use of UV lamps may be detectable depending on the deployment environment.
2. The paper could further discuss potential defenses.

---

> ### Author Rebuttal · Authors · 2025-07-30
>
> **We sincerely thank you for your insightful and positive feedback!**
>
> **Q-1. UV light is invisible, using a UV lamp may be visible. How can the attacker ensure that the UV lamp remains covert? Moreover, in some scenarios, such as highways, where traffic signs are usually placed high, can the attack still be successful at this point? It would be helpful if this aspect could be explained to alleviate the concerns about the stealthiness.**
>
> A-1. We tested the impact of distance between UV lamp and traffic sign in **Appendix Section D** and found that there is still an attack success rate of at least 81% at a distance of 8 meters. This distance is sufficient for an attacker to place the UV lamp in the bushes near the traffic sign. In addition, since the UV light is invisible to the naked eye, the victim cannot locate the light source, and the attack device is in the bushes, further increasing the stealthiness of the attack. For elevated traffic signs commonly found on highways, attackers can leverage drones to carry out the attack. By equipping a drone with a UV lamp and hovering it at a specific position, the attacker can activate the perturbation without physical access to the sign. The drone only needs to remain in position for a few seconds during the attack and can quickly leave the area afterward, making the attack highly stealthy and flexible.
>
> **Q-2. The paper should further discuss the feasibility of defenses, e.g., could a UV filter defend against the attacks in this paper? Relying solely on high-definition maps may not be sufficient, especially in cases where traffic signs have been replaced by workers but the maps are not yet updated.**
>
> A-2. A UV filter **cannot defend** against FIPatch. First, they cannot stop fluorescent materials from absorbing UV light, and the emitted visible light is still captured by cameras. Second, UV filters hide the attack source by blocking UV light in images. Apart from high-definition maps, we believe that collaborative perception among multiple vehicles is also a promising defense strategy. Specifically, while an attacker may launch adversarial attacks using fluorescent ink against a specific vehicle, other unaffected vehicles are still able to recognize the traffic sign correctly. By leveraging recognition results from nearby vehicles on the same road, the victim’s vehicle can correct its prediction. This cross-vehicle information sharing mechanism can significantly enhance the system’s robustness against adversarial attacks.
>
> **Q-3. Although physical experiments are thorough, the impact is only measured at the AI component level (classification/detection), not on full AV systems (e.g., driving decisions).**
>
> A-3. Our paper has mentioned this issue in the limitation. For fairness, we followed prior studies [1]-(USENIX-2023), [2]-(NDSS-2022), [3]-(USENIX-2021), and mentioned the AD systems in limitation. If insist, we could provide results on real-AD systems.
>
> **Reference**
>
> [1] Wenjun Zhu, Xiaoyu Ji, Yushi Cheng, Shibo Zhang, and Wenyuan Xu. Tpatch: A triggered physical adversarial patch. In *USENIX Security*, 2023.
>
> [2] Wei Jia, Zhaojun Lu, Haichun Zhang, Zhenglin Liu, Jie Wang, and Gang Qu. Fooling the eyes of autonomous vehicles: Robust physical adversarial examples against traffic sign recognition systems. *NDSS*, 2022.
>
> [3] Giulio Lovisotto, Henry Turner, Ivo Sluganovic, Martin Strohmeier, and Ivan Martinovic. SLAP: Improving physical adversarial examples with Short-Lived adversarial perturbations. In *USENIX Security*, 2021.

---

> > ### Comment · Reviewer_7D6S · 2025-08-07
> >
> > Thanks for your response. I have read the authors’ rebuttal and the comments from other reviewers. I appreciate the additional experimental results and the discussion addressing my concerns, e.g., stealthiness and potential defenses. I believe that fluorescent ink is a novel physical medium for adversarial attacks. The experimental results, particularly those in the physical world, demonstrate the effectiveness and robustness of the proposed approach. Therefore, I will maintain my score and support the acceptance of this paper.

---

### Official Review · Reviewer_MB7X · 2025-06-24

**Clarity:** 2
**Significance:** 3
**Originality:** 4
**Rating:** 3
**Confidence:** 4

**Summary:**

This paper introduces a new physical attack vector that leverages fluorescent ink to craft adversarial patches targeting traffic sign recognition models. The physical realization involves the use of invisible UV lighting, which activates the otherwise invisible fluorescent ink, rendering the attack patch visible only under specific conditions. This enables attackers to dynamically control the activation of the patch, offering a significant advantage over traditional passive sticker-based attacks. The effectiveness of the proposed method is evaluated across multiple datasets and model architectures.

**Questions:**

Questions
1. This work seems more focused on engineering. Aside from the patch fabrication, what are the new technical or methodological contributions?
2. There are existing physical attack baselines (e.g., sticker, projector). Can the authors include these for a fair comparison?
3. The attack success rates are very high on the authors' own setting, which suggests possible overfitting. It's unclear how well the method generalizes to different conditions (e.g., new environments, bad weather).
4. The supplementary images are low resolution (32×32). Was the patch really designed at this size and used directly in the physical setup? If so, more explanation is needed.

Suggestions
1. Moving the method details (L60–70) from the Introduction to the Methods section would improve the structure and clarity of the paper.
2. The citation style should be revised to follow a standard academic formatting convention.

**Ethical Concerns:**

["NO or VERY MINOR ethics concerns only"]

**Final Justification:**

Most of my concerns have been addressed throughout the discussion. However, the current submission may still require major revisions to incorporate the provided responses and present the work more clearly. As a result, I lean toward rejecting the paper, though I believe the revised version could be acceptable for submission to a next top-tier venue.

**Limitations:**

The fluorescent patch is too visible, which may limit its practicality in real-world use. It also requires an additional UV lamp, adding extra cost and complexity.

**Quality:**

1

**Strengths And Weaknesses:**

Strengths
- The idea of leveraging fluorescent ink for physical adversarial attacks is quite interesting and plausible.
- The paper includes various evaluations and ablation analyses.

Weaknesses
- The paper shows solid engineering work in using fluorescent ink for physical adversarial patches, but it lacks new methodological contributions. It mainly applies existing physical attack techniques to a new material.
- The methodology section needs more detail and clearer justification, especially for the physical implementation.
- The fluorescent patch is too visible, which may reduce its practicality in real-world scenarios.
- The evaluation uses outdated CNN models. It should include newer models like vision transformers.
- The high attack success rates (98.31% and 96.72%) suggest the evaluation doesn't consider real-world challenges like out-of-distribution data or adverse weather.
- The paper only tests defenses designed for digital attacks. It should include evaluations with more advanced physical defenses.
- The writing needs improvement for clarity and readability.

Typos
- L27: focue -> focus
- L35: Sato et al. (Redundant)
- L38: Zhu et al. (Redundant)
- L118: Zhu et al. (Redundant)

---

> ### Author Rebuttal · Authors · 2025-07-30
>
> **Q-1. Lacks new methodological contributions.**
>
> A-1. We highlight again that we explore a new adversarial attack **strategy** by using the fluorescent ink to achieve a more stealthy and effective attack in the physical world. Compared with existing methods, our FIPatch offers the following advantages:
>
> 1. High effectiveness: Existing methods are often vulnerable to defense strategies, such as optical filters or physical signal protection mechanisms. In contrast, our approach cannot be easily defended against by current techniques and demonstrates strong robustness in the physical world. This is validated by both digital and physical experiments.
>
> 2. Unconspicuous: Our method supports flexible triggering. It remains invisible when inactive and minimizes the perturbation area when activated. In the physical world, the perturbation only needs to be active for a short time. When a self-driving car approaches the stop sign, even if it fails to recognize the stop sign for merely a short time window, it can lead to a fatal accident.
>
> Although fluorescent ink enables invisible and actively triggered features, designing an effective and robust physical adversarial patch introduces **new** challenges.
>
> (1) It is challenging to simulate fluorescent effects and determine the optimal choices of massive factors in fluorescent ink, e.g., color, transparency, and size, for achieving high attack effectiveness.
>
> (2) Fluorescent effects are easily influenced by real-world environments, including surroundings, ambient light, and vehicle distance.
>
> **Existing methods do not address these challenges**, and most require manual annotation of traffic signs. To address the challenge (1), we first develop a color-edge fusion method to automatically locate traffic signs, enabling precise application of fluorescent ink to the signs, rather than invalid backgrounds. Second, to effectively simulate fluorescent effects, we model fluorescent perturbations on traffic signs by defining the various critical parameters of fluorescent ink, including colors, intensities, and perturbation sizes. Third, we design goal-based and area-based loss functions to achieve high ASRs with minimal perturbations, supporting three attack goals: hiding attack, generative attack, and misclassification attack. To solve the challenge (2), we present several fluorescence-specific transformation methods to improve the robustness of FIPatch in the physical world. The consistency between digital and physical experimental results further validates the effectiveness of our approach.
>
> We still believe our method is novel. In the next version, we will extend the contributions to elaborate on our innovations and highlight the differences from existing methods.
>
> **Q-2. The fluorescent patch is visible.**
>
> A-2. Please refer to **Reviewer 1t4h’s Q-3**.
>
> **Q-3. The evaluation uses outdated CNN models.**
>
> A-3. We follow prior work (e.g., [1] (USENIX-2023), [2] (CVPR-2022), [3] (NDSS-2022), [4] (USENIX-2021), [5] (CVPR-2021), [6] (NDSS 2024)) by using both one-/two-stage detectors and classifiers (including CNN) for TSR.
>
> **Q-4. Can the authors include existing physical attacks (e.g., sticker, projector) for a fair comparison?**
>
> A-4. It is **unfair** to compare existing work in the experimental section. Due to the diversity of physical environments, the difficulty of acquiring attack equipment, and the unavailability of source code, most physical attacks [6] (NDSS 2024), [1] (USENIX-2023), [2] (CVPR-2022), [4] (USENIX-2021), [5] (CVPR-2021) are compared with other methods only from a theoretical perspective. In addition, sticker attacks all vehicles indiscriminately once deployed, which lacks flexibility. Although projectors can be actively triggered, the cost of attack is too high, and the price of projectors used in [4] ranges from $1,500 to $44,379. Therefore, we follow the comparison approach adopted in these works.
>
> To address the reviewer’s concern, we establish several baselines for comparison:
>
> - For the hiding attack, we cover the entire surface of the traffic sign (TS) with fluorescent ink.
> - For the misrecognition attack, we draw a 14cm-radius fluorescent ink circle at the center of a 60 cm × 60 cm TS.
> - For the generative attack, a naive approach is to imitate a real TS. However, this compromises stealth, as humans can easily recognize the fake sign.
>
> The ASR results are presented below:
>
> |          Scheme           | Yolov3 | Faster R-CNN | CNN  | VGG13 |
> | :-----------------------: | :----: | :----------: | :--: | :---: |
> |     Baseline (Hiding)     |   3%   |      1%      |  -   |   -   |
> |          FIPatch          |  91%   |     88%      |  -   |   -   |
> | Baseline (Misrecognition) |  17%   |     13%      | 24%  |  19%  |
> |          FIPatch          |  100%  |     98%      | 100% |  99%  |
>
> The results show that the baseline methods yield low ASR, with none exceeding 24%. This proves that such naive approaches are invalid and validates the effectiveness of our method. In the next version, we will include the baseline comparison. If insist, we will also incorporate results from related work for further comparison.
>
> **Q-5. The high ASR suggests possible overfitting. How well the method generalizes to different conditions?**
>
> A-5. A high ASR does not imply that real-world challenges were overlooked. We conducted experiments in both digital and physical settings, along with extensive ablation studies. Especially in the physical-world experiments, FIPatch consistently achieves high ASR across varying distances, angles, UV lamp powers, and vehicle speeds. In fact, the consistency between our digital and physical experiments validates the effectiveness of our approach. Following your suggestion, we evaluated FIPatch under rainy conditions by simulating rain with a sprinkler. The ASR dropped by only 1%. Additional weather experiments will be included in the next version.
>
> Moreover, prior works have also achieved high ASR. For example, [1] (USENIX-2023) reported an ASR of 96%, and [6] (NDSS 2024) achieved 100%. Therefore, our method **does not suffer from overfitting**.
>
> **Q-6. It should include evaluations with more advanced physical defenses.**
>
> A-6. We sincerely appreciate your insightful suggestion! We consider the following physical defenses: camera-LiDAR fusion, optical filters, and multi-camera fusion.
>
> - Camera-LiDAR fusion: Since traffic signs are planar objects, LiDAR cannot capture the content of the sign. As a result, existing TSR systems primarily rely on cameras.
>
> - Optical filters: Our attack remains effective even with optical filters. Since fluorescence emits visible light, only visible-light filters could block it. However, no cameras in current AVs use visible-light filters. Moreover, filtering a single color does not prevent the effectiveness of fluorescent materials in other colors.
>
> - Multi-camera fusion: We evaluated multi-sensor fusion defenses in the physical world. Specifically, we captured the same traffic sign from five different angles (−20°, −10°, 0°, 10°, and 20°) and applied a voting-based decision strategy. The ASR results are shown below:
>
> |       Scheme        | CNN  | ResNet50 | VGG13 | GoogleNet |
> | :-----------------: | :--: | :------: | :---: | :-------: |
> |     W/o defense     | 100% |   99%    |  99%  |    99%    |
> | Multi-sensor fusion | 97%  |   94%    |  95%  |    95%    |
>
> Experimental results show that multi-sensor fusion can reduce the ASR by at most 5%, demonstrating the strong robustness of FIPatch. Moreover, as illustrated in Figure 5, ASR is less sensitive to changes in angle but is significantly affected by variations in distance. In the next version, we will evaluate additional physical-world defense strategies and provide further discussion on the results.
>
> **Q-7. The supplementary images are low resolution (32×32). Was the patch really designed at this size?**
>
> A-7. We apologize for the possible misunderstanding about the experimental setup. **Our FIPatch is designed for the physical setup.** The 32×32 size refers to the image’s resolution on the dataset. In the physical setup, the traffic sign is 60cm×60cm, and the real-world adversarial patch, typically no larger than 14cm, is resized and adjusted using EOT transformations to ensure robustness under various physical conditions. In the next version, we will provide a more detailed explanation of the settings used in the physical world.
>
> **Q-8. Moving the method details (L60–70) to the Methods section.**
>
> A-8. Thanks for your valuable suggestion! In the next version, we will move the details to the methodology section, especially for the physical implementation.
>
> **Reference**
>
> [1] Wenjun Zhu, Xiaoyu Ji, Yushi Cheng, Shibo Zhang, and Wenyuan Xu. Tpatch: A triggered physical adversarial patch. In *USENIX Security*, 2023.
>
> [2] Yiqi Zhong, Xianming Liu, Deming Zhai, Junjun Jiang, and Xiangyang Ji. Shadows can be dangerous: Stealthy and effective physical-world adversarial attack by natural phenomenon. In *CVPR*, 2022.
>
> [3] Wei Jia, Zhaojun Lu, Haichun Zhang, Zhenglin Liu, Jie Wang, and Gang Qu. Fooling the eyes of autonomous vehicles: Robust physical adversarial examples against traffic sign recognition systems. In *NDSS*, 2022.
>
> [4] Giulio Lovisotto, Henry Turner, Ivo Sluganovic, Martin Strohmeier, and Ivan Martinovic. SLAP: Improving physical adversarial examples with Short-Lived adversarial perturbations. In *USENIX Security*, 2021.
>
> [5] Alon Zolfi, Moshe Kravchik, Yuval Elovici, and Asaf Shabtai. The translucent patch: A physical and universal attack on object detectors. In *CVPR*, 2021.
>
> [6] Takami Sato, Sri Hrushikesh Varma Bhupathiraju, Michael Clifford, Takeshi Sugawara, Qi Alfred Chen, and Sara Rampazzi. Invisible reflections: Leveraging infrared laser reflections to target traffic sign perception. In *NDSS*, 2024.
>
> [7] Eykholt K, Evtimov I, Fernandes E, et al. Robust physical-world attacks on deep learning visual classification. In *CVPR*, 2018.

---

> > ### Comment · Reviewer_MB7X · 2025-08-05
> >
> > I think the authors for their detailed responses on the rebuttal. Some of my concerns (visibility of the patch, low resolution) have been resolved, but several concerns remain. I provide the remaining concerns below:
> >
> > ---
> > **Q-1. Lacks new methodological contributions.**
> >
> > **Q-6. It should include evaluations with more advanced physical defenses.**
> >
> > The provided responses do not address the methodological novelty or contributions but instead focus on the strengths of fluorescent material-based attacks. First, 'high effectiveness' is achieved through simple baseline attack losses with EoT-based robustness. The authors do not provide concrete theoretical or empirical evidence supporting the claim that 'existing methods are often vulnerable to defense strategies, while our approach cannot be easily defended.' Defense evaluations should be conducted adaptively, not on existing baselines that are not relevant to the fluorescent modality. However, the reviewer acknowledges the unique considerations of fluorescent effects, which distinguish this approach from previous methods.
> >
> > **Q-3. The evaluation uses outdated CNN models.**
> >
> > Although prior work focuses solely on CNN models, evaluating more recent CNN and ViT-based models would strengthen the claim, as physical attacks represent the frontier of real-world threats. Therefore, the victim models used in the evaluation should align with the current state-of-the-art to reflect true security aspects.
> >
> > **Q-4. Can the authors include existing physical attacks (e.g., sticker, projector) for a fair comparison?**
> >
> > To clarify the misunderstanding, the reviewer does not require the proposed method to outperform all existing types of physical attacks. A direct comparison is unfair, as the targeted attack vectors differ across methods and materials. However, to better position the proposed methodology, it is highly encouraged to compare the methods in mostly controlled environments (e.g., comparing the effectiveness of projector-based, sticker-based, and fluorescent-based methods under the same perturbation area). We appreciate the example baselines provided, but the comparison should focus on prior competitive methods rather than those that are insightfully less effective.
> >
> > **Q-5. The high ASR suggests possible overfitting. How well the method generalizes to different conditions?**
> >
> > The reviewer’s suggestion that an excessively high ASR may indicate possible overfitting could be controversial; however, the reviewer still believes that an ASR above 95% in every setting implies a flawed evaluation. As this work targets physical attacks in real-world settings, more rigorous evaluations under diverse adverse weather conditions should be considered. We appreciate the additional experiments conducted under rainy conditions, but it is difficult to substantiate the claim without further details on the experiments. Additionally, evaluations in various out-of-distribution scenarios within a simulated environment could further strengthen the effectiveness of the proposed method.
> >
> > ---
> > The reviewer would appreciate it if other reviewers who made positive remarks could also address the concerns outlined above.

---

> > > ### Author Response · Authors · 2025-08-07
> > >
> > > **We sincerely thank the reviewer for the detailed and constructive feedback. We appreciate the thoughtful comments and have carefully addressed each concern below.**
> > >
> > > **Q-1. The provided responses do not address the methodological novelty or contributions.**
> > >
> > > A-1. Thanks for recognizing the unique consideration of fluorescence effects that distinguishes this approach from previous methods. We would like to clarify that our contribution goes beyond simply introducing a new material (fluorescent ink). Instead, we build a **systematic attack optimization framework** tailored to the physical properties of fluorescent ink. Specifically:
> > >
> > > - In Section 4.1, we propose an automatic localization method to ensure **consistency** between the location of the perturbation in the digital and physical environments. This **region-aware strategy**, which combines automatic localization with multi-objective optimization, represents a novel approach compared to existing methods that rely on manual annotation.
> > > - In Section 4.2, we define and systematically model several key parameters related to fluorescent ink, including position, radius, color, and intensity. To the best of our knowledge, this is the first attempt to **parameterize fluorescent perturbations** for attack optimization.
> > > - In Section 4.3, we additionally design a novel **area-based** loss function to support three types of attack objectives.
> > > - In Section 4.4, improving robustness does not rely solely on existing EoTs. We **extend the distribution** of EoT transformations based on the properties of fluorescent materials and introduce a **transparency transformation method** to better simulate real-world variations.
> > >
> > > The combination of these components constitutes the **methodological novelty** of our work, beyond just engineering implementation. In the next revision, we will highlight these technical innovations to better express the methodological contributions of our approach.
> > >
> > > **Q-2. The authors do not provide concrete theoretical or empirical evidence supporting the claim that 'existing methods are often vulnerable to defense strategies, while our approach cannot be easily defended.' Defense evaluations should be conducted adaptively.**
> > >
> > > A-2. We first explain why existing methods are vulnerable to defense.
> > >
> > > - Projector-based attacks[1] can be easily traced back to the **visible light** source, making them insufficiently stealthy in real-world scenarios. Our method uses **invisible** UV light to activate the fluorescent ink, which cannot be visually detected or traced to the attack source.
> > > - Infrared (IR)-based attacks [2] are vulnerable to simple physical defenses, such as **IR filters**. As highlighted in [2], “recommended that CAV companies use IR filters” to effectively block such attacks. Our attack cannot be defended against by **any** existing optical filter.
> > > - Acoustic signal-based attacks [3] can be easily blocked by **physical signal protection** mechanisms [4] [5]. Our method is purely **vision-based** and leverages the **camera input pipeline**.
> > >
> > > Regarding adaptive defenses, unfortunately, we have not yet found any effective strategies specifically targeting fluorescent materials. One might consider using UV filters (an idea inspired by [2]) since the attacker relies on ultraviolet light. However, a UV filter in the camera does not block fluorescent-light absorption, and the visible light is still captured while hiding the attack source.
> > >
> > > We will include these detailed explanations in the revision to help readers better understand the advantages of our approach.
> > >
> > > **Due to the word limit, we addressed Q-1 and Q-2 in this comment. Responses to Q-3–Q-5 are provided in the next one.**

---

> > > ### Author Response · Authors · 2025-08-07
> > >
> > > **Q-3. The victim models used in the evaluation should align with the current state-of-the-art to reflect true security aspects.**
> > >
> > > A-3. We additionally include a Vision Transformer (ViT) model in the traffic sign classification task. Specifically, we use the `ViT-B_16` variant, loaded via the `vision_transformer_pytorch` package with pretrained weights. The model is trained to classify traffic signs in the GTSRB dataset. We use cross-entropy loss as the objective function and optimize the model using the AdamW optimizer with a learning rate of `1e-4` and a weight decay of `0.001`. A StepLR scheduler is applied with a step size of 2 and a decay factor (`gamma`) of 0.1 to adjust the learning rate during training.
> > >
> > > We evaluate FIPatch using the ViT model in the physical world. We record 3000 frames under various lighting conditions, and other experimental settings are the same as in Section 5.2 (e.g., the distance between the vehicle and the sign is 5 m, the angle was fixed at 0°, and the power of the UV lamp is 120W). The ASR results are shown below:
> > >
> > > | Model/Light | 200 lux | 500 lux | 1000 lux |
> > > | :---------: | :-----: | :-----: | :------: |
> > > |     ViT     | 99.43%  | 98.63%  |  95.17%  |
> > >
> > > The results demonstrate that FIPatch also remains effective against the ViT model under various illumination conditions. We will include the above results in the revision.
> > >
> > > **Q-4. To better position the proposed methodology, it is highly encouraged to compare the methods in mostly controlled environments (e.g., comparing the effectiveness of projector-based, sticker-based, and fluorescent-based methods under the same perturbation area).**
> > >
> > > A-4. We appreciate your insightful suggestion. Following your recommendation, we conducted additional comparisons with representative sticker-based [6] and projector-based [1] physical attack methods, using the publicly available codes. All experiments were conducted under consistent physical conditions: the ambient light was 500 lux, the distance between the vehicle and the sign was 5 m, and the angle was fixed at 0°. We used a CNN model (a publicly available implementation of a multi-scale CNN architecture that has been known to perform well on road sign recognition)  trained on the GTSRB dataset and performed misclassification attacks.
> > >
> > > - **Projector-based attack setup**: We use a Sony VPL-EX450, a mid-range office projector with a maximum brightness of 3,600 lumens. The projector is placed 2 meters away from the TS. To ensure accurate projection, we apply a homography transformation to geometrically align the projected perturbation with the contour of the TS.
> > > - **Sticker-based attack setup**: Following the optimization strategy in [6], we use the Adam optimizer with the following parameters: $\beta_1 = 0.9,\ \beta_2 = 0.999,\ \epsilon = 10^{-8}$, and a learning rate $\eta \in [10^{-4}, 10^{0}]$. As suggested in [6], for untargeted attacks, we modify the objective function to maximize the distance between the model’s prediction and the true class.
> > >
> > > Experimental results are shown below:
> > >
> > > | Metric/Scheme | Sticker | Projector | FIPatch |
> > > | :-----------: | :-----: | :-------: | :-----: |
> > > |      ASR      | 91.03%  |  97.47%   | 98.83%  |
> > >
> > > We appreciate your agreement that FIPatch does not need to outperform all existing physical attacks in terms of ASR. Following your suggestion, we will include this table in the revision.

---

> > > ### Author Response · Authors · 2025-08-07
> > >
> > > **Q-5. An ASR above 95% in every setting implies a flawed evaluation. As this work targets physical attacks in real-world settings, more rigorous evaluations under diverse adverse weather conditions should be considered.**
> > >
> > > A-5. Many existing physical-world attacks have achieved ASRs above 95%. For example, [7] (NDSS 2025) and [2] (NDSS 2024) both achieved a 100% ASR. [3] (USENIX 2023) reported an ASR of 96%, while [8] (CVPR 2022) and [1] (USENIX 2021) also achieved up to 100% ASR. Unfortunately, none of the above works considered the impact of weather conditions in their experiments. This is likely because the weather is uncontrollable and difficult to quantify.
> > >
> > > Our physical-world experiments have taken into account various real-world factors, including lighting conditions, distance, angle, and vehicle speed. To simulate rainy conditions, we conduct experiments under a lighting intensity of 1000 lux, with the vehicle positioned 5 meters away from the traffic sign and the UV lamp set to 120W. In the dry setting, we turn off the sprinkler and record 3,000 image frames to evaluate the ASR of FIPatch. In the rainy setting, we activate the sprinkler to spray water onto the traffic sign while recording another 3,000 frames. This allowed us to assess the impact of rain on FIPatch’s performance. The results show that rain caused only a minor drop in ASR, approximately 1%.
> > >
> > > As for more extreme weather conditions (e.g., heavy rain or dense fog), such environments can potentially reduce the visibility of traffic signs. Even without any adversarial attack, models may struggle to recognize targets under such conditions. Therefore, it becomes impractical to evaluate attack effectiveness in these settings, as the model’s baseline performance is already impaired. We appreciate your suggestion and will include a discussion on weather conditions in the revision.
> > >
> > >
> > >
> > > **We sincerely appreciate your valuable suggestions. We believe these issues can be addressed in the revision. Please let us know if you have any further questions.**
> > >
> > >
> > >
> > > **Reference**
> > >
> > > [1] Giulio Lovisotto, Henry Turner, Ivo Sluganovic, Martin Strohmeier, and Ivan Martinovic. SLAP: Improving physical adversarial examples with Short-Lived adversarial perturbations. In *USENIX Security*, 2021.
> > >
> > > [2] Takami Sato, Sri Hrushikesh Varma Bhupathiraju, Michael Clifford, Takeshi Sugawara, Qi Alfred Chen, and Sara Rampazzi. Invisible reflections: Leveraging infrared laser reflections to target traffic sign perception. In *NDSS*, 2024.
> > >
> > > [3] Wenjun Zhu, Xiaoyu Ji, Yushi Cheng, Shibo Zhang, and Wenyuan Xu. Tpatch: A triggered physical adversarial patch. In *USENIX Security*, 2023.
> > >
> > > [4] Ramesh C Sharma, Subodh Kumar, Surya Kumar Gautam, Saurabh Gupta, Deepak Kumar, and Hari B Srivastava. Detection of ultrasonic waves using resonant cylindrical cavity for defense application. *IEEE Sensors Journal*, 2017.
> > >
> > > [5] Jianzhi Lou, Qiben Yan, Qing Hui, and Huacheng Zeng. Soundfence: Securing ultrasonic sensors in vehicles using physical-layer defense. *IEEE SECON*, 2021.
> > >
> > > [6] Kevin Eykholt, Ivan Evtimov, Earlence Fernandes, Bo Li, Amir Rahmati, Chaowei Xiao, Atul Prakash, Tadayoshi Kohno, Dawn Song. Robust Physical-World Attacks on Deep Learning Visual Classification. In *CVPR*, 2018.
> > >
> > > [7] Takami Sato, Ryo Suzuki, Yuki Hayakawa, Kazuma Ikeda, Ozora Sako, Rokuto Nagata, Ryo Yoshida, Qi Alfred Chen, Kentaro Yoshioka. On the Realism of LiDAR Spoofing Attacks against Autonomous Driving Vehicle at High Speed and Long Distance. In *NDSS*, 2025.
> > >
> > > [8] Yiqi Zhong, Xianming Liu, Deming Zhai, Junjun Jiang, and Xiangyang Ji. Shadows can be dangerous: Stealthy and effective physical-world adversarial attack by natural phenomenon. In *CVPR*, 2022.

---

> > > > ### Comment · Reviewer_MB7X · 2025-08-07
> > > >
> > > > I greatly appreciate the authors for their rigorous efforts and additional experimental validations. Their responses have significantly helped me understand the novelty and contributions of the work. Most of my concerns have been addressed, and I believe these improvements will benefit the quality of the paper. As a result, I have increased my score. However, as indicated in my initial rating, the current submission may still require major revisions to sufficiently address its methodological and experimental details, ensuring a reproducible foundation for physical attacks.

---

### Official Review · Reviewer_1t4h · 2025-07-02

**Clarity:** 3
**Significance:** 3
**Originality:** 2
**Rating:** 4
**Confidence:** 4

**Summary:**

This paper introduces FIPatch, a stealthy physical-world adversarial attack that uses fluorescent ink to alter the appearance of traffic signs.

**Questions:**

1 As shown in Figure 15, the central region of the image exhibits the highest percentage of successful attacks, indicating that attacks targeting the center of the sign are more effective. Given that the fluorescent ink can only be applied on the traffic sign itself, I believe the objective function defined in your formulation (Eq. 8) is already capable of implicitly optimizing the patch placement toward the most effective region, typically near the center of the sign. Therefore, the method introduced in Section 4.1 for region selection feels unnecessary or redundant. If this step is essential, the paper should better justify its inclusion and clearly explain how it improves upon direct optimization via the objective function alone.
2 In Section 5, Table 2, it’s unclear whether the attack is white-box or black-box. If it’s white-box, this may not reflect realistic real-world scenarios. Please clarify the threat model and consider evaluating the attack in a black-box transfer setting to better demonstrate its practical robustness.
3 The authors should consider imposing a hard constraint on the attack area size to ensure the perturbation remains imperceptible to human drivers. As shown in Figure 4, the first two traffic signs are difficult to recognize even for humans. If successful attacks require such visually obtrusive modifications, similar effects could be achieved by simply occluding or blocking the sign, which weakens the motivation for an adversarial approach.

**Ethical Concerns:**

["NO or VERY MINOR ethics concerns only"]

**Final Justification:**

Most of the issues I raised have been adequately addressed by the authors, so I am updating my score accordingly. I suggest revising the remaining unclear part for clarity.

**Limitations:**

See questions

**Quality:**

3

**Strengths And Weaknesses:**

The authors conduct extensive experiments in both digital and physical environments, covering three distinct attack goals

---

> ### Author Rebuttal · Authors · 2025-07-30
>
> **Thanks for your valuable comment. We have explained your questions and modified the corresponding content in the next version according to your suggestions.**
>
> **Q-1. The method introduced in Section 4.1 for region selection feels unnecessary or redundant.**
>
> A-1. We apologize for the possible misunderstanding about Section 4.1. We would like to emphasize that Section 4.1 is indispensable for the following two reasons:
>
> - **The final perturbation may extend beyond the region of the traffic sign (TS).** For example, a real-world image captured by a camera typically includes both the TS and its background. Direct optimization in such cases may generate perturbations that spread into the background, which cannot be physically realized using fluorescent ink. Additionally, although Figure 15 shows that the central region of the TS exhibits the highest percentage of successful attacks, perturbations with a large radius may still exceed the TS boundary. Therefore, it is essential to apply the mask defined in Section 4.1 to ensure that the perturbation remains strictly within the TS area.
> - **Section 4.1 significantly benefits the optimization process.** In real-world scenarios, it is impractical to manually test a wide range of attack parameters on physical TSs. Thus, we propose a simulation-and-optimization framework to search for optimal attack parameters. The automatic localization in Section 4.1 enables a fully automated attack pipeline, ensuring that the generated perturbation is both deployable and confined to the TS region. As a result, attackers can directly apply the optimized perturbation in the physical world with high attack success rates (ASRs).
>
> We will revise the manuscript to reduce the redundancy and better justify the necessity of Section 4.1 and clearly explain how it improves the attack pipeline.
>
> **Q-2. Please clarify the threat model and consider evaluating the attack in a black-box transfer setting to better demonstrate its practical robustness.**
>
> A-2. **We already evaluated FIPatch’s transferability in Table 5 of the Appendix**. We define our attack as a black-box setting in Section 3.2, where the attacker has no access to the internal parameters, architecture, or gradients of the target model. This black-box setting is highly practical in the physical world. As shown in Table 5, the ASRs exceed 85% when transferring to models such as ResNet50, ResNet101, VGG13, and VGG16. The lowest ASR, 69%, occurs when transferring from Inception v3 to MobileNet v2. High ASRs, such as the 95% observed when transferring from ResNet50 to ResNet101, indicate that models with similar architectures are particularly vulnerable. These experimental results demonstrate the robustness of FIPatch across different models. In the next version, we will move these transferability experiments to Section 5 and highlight the black-box setting in the threat model.
>
> **Q-3. The authors should consider imposing a hard constraint on the attack area size to ensure the perturbation remains imperceptible to human drivers. As shown in Figure 4, the first two traffic signs are difficult to recognize even for humans.**
>
> A-3. In our design (Section 4.3), we have already considered an area-based loss, which introduces the smallest perturbation, to enhance the practicality of FIPatch in the physical world. As shown in Figure 4, our generative attack can fool the TSR system even when the perturbation is only a ring, and a driver will not recognize it as a "STOP" sign. In both Hiding and Misrecognition attacks, the driver can still recognize the original traffic sign. Moreover, for the hiding attack, when the attacker does not trigger the perturbation, the traffic sign still appears as a normal stop sign to human drivers; however, once triggered, the TSR system fails to detect it. It is worth noting that for the Hiding Attack, the perturbation radius tends to be larger. This may be because larger perturbations are required to shift the model's attention away from the original traffic sign. We will clarify and discuss this observation in the next version.

---

> ### Comment · Reviewer_1t4h · 2025-08-05
>
> Q1. According to Figure 15, the attack success rate is highest in the central region of the traffic sign (TS), while other regions perform significantly worse. This suggests that the optimization process, if unconstrained, would naturally converge toward placing perturbations near the center of the TS to achieve a successful attack. Given this, I wonder whether the optimization alone — without the region localization step in Section 4.1 — might already be sufficient to discover effective perturbation locations. Have you tried running the optimization without the localization module and evaluated how well the adversarial attacks perform in that case? A comparison would help clarify whether Section 4.1 offers substantial additional benefits over direct optimization.
>
> Q2 I noted that the black-box transferability results are included in Table 5 of the appendix. However, I noticed that Table 5 reports only simulation results, whereas Table 2 presents physical-world performance. My original concern was more specifically about evaluating the black-box setting in the physical world, since that’s ultimately the scenario the attack aims to succeed in. Currently, Table 2 seems to reflect a white-box physical setup.
>
> Q3. Thank you for the explanation. I appreciate that the area-based loss in Section 4.3 is intended to limit the size of the perturbation. However, my concern is specifically about visual imperceptibility to human drivers, not just minimizing the perturbation area. Some of the perturbations shown in Figure 4 still appear quite obtrusive, and in those cases, the attack could arguably be replicated by simpler physical occlusion or vandalism, which weakens the motivation for using adversarial perturbations specifically. Have you considered adding a hard constraint on perturbation size (or a perceptual similarity constraint) to enforce a stronger realism requirement?

---

> > ### Author Response · Authors · 2025-08-06
> >
> > **We sincerely appreciate your thoughtful feedback! We have provided detailed responses to your concerns, and please let us know if you'd like more adjustments.**
> >
> > **Q-1. I wonder whether the optimization alone — without the region localization step in Section 4.1 — might already be sufficient to discover effective perturbation locations.**
> >
> > A-1. We apologize for the confusion caused by Figure 15. The results in Figure 15 demonstrate that **when the optimization process is constrained by Section 4.1 (rather than unconstrained)**, the perturbation tends to be placed near the center of the traffic sign. Without the localization module introduced in Section 4.1, it would be impossible to deploy the attack in the physical world. Specifically, if the location of the perturbation is not restricted, the model can still be misled even when the perturbation lies outside the traffic sign region (this was a problem we identified during the early stages of our design). Moreover, even if the center of the perturbation lies on the traffic sign, the circle may still extend beyond the sign’s boundary, which can lead to failure in physical deployment. In summary, **the purpose of Section 4.1 is not to significantly improve the ASR**, but to ensure that the attack can be physically deployed and fully automated.
> >
> > **Q-2. My original concern was more specifically about evaluating the black-box setting in the physical world, since that’s ultimately the scenario the attack aims to succeed in. Currently, Table 2 seems to reflect a white-box physical setup.**
> >
> > A-2. Similar to prior works [1] [2] [3], we adopt a black-box setting for both digital and physical experiments. In this context, “black-box” means that the attacker has no access to the internal parameters or gradients of the model and can only query its outputs during the attack process. The results in Table 2 were obtained under the black-box setting (not white-box), where only the model’s outputs were used during the attack. The experimental results indicate that our method achieves competitive performance in the physical world compared to its performance in the digital domain (consistency between the digital and physical results). If insist, we will include evaluations of transferability in the physical world in the revision (within 7 days).
> >
> > **Q-3. My concern is specifically about visual imperceptibility to human drivers, not just minimizing the perturbation area. Have you considered adding a hard constraint on perturbation size (or a perceptual similarity constraint) to enforce a stronger realism requirement?**
> >
> > A-3. **Yes**, our approach incorporates a hard constraint on the size of the perturbation. As described in Section 4.2, we set an upper bound $r_{max}$ on the radius of the perturbation circle. During optimization, the radius $r$ is constrained within the range $[r_{min}, r_{max}]$.
> >
> > Moreover, visual imperceptibility to human drivers is highly subjective, so we attempt to quantify it using the relative area $R$ = (perturbation area / traffic sign area). In Figure 13 of the appendix, we analyze how the maximum radius affects ASR. When $r_{max}=3$, the ASR already exceeds 90%. Since the image size is 32×32 and the traffic sign (a circle) is assumed to occupy the entire image, the maximum relative area $R$ is approximately 3/16=18.75%. In practice, the optimized radius is often smaller than $r_{max}$, resulting in a smaller $R$. Furthermore, as shown in Figure 12, the fluorescent material becomes nearly invisible after being left for five days, meaning the patch remains unnoticeable when the attack is not triggered. This significantly enhances the stealth and flexibility of the attack.
> >
> > Finally, we emphasize that in the physical world, **the perturbation only needs to be active for a short time**. When a self-driving car approaches the stop sign, even if it fails to recognize the stop sign for merely a short time window, it can lead to a fatal accident.
> >
> >
> >
> > [1] Daizong Ding, Mi Zhang, Fuli Feng, Yuanmin Huang, Erling Jiang, Min Yang. Black-Box Adversarial Attack on Time Series Classification. In *AAAI*, 2023.
> >
> > [2] Takami Sato, Sri Hrushikesh Varma Bhupathiraju, Michael Clifford, Takeshi Sugawara, Qi Alfred Chen, and Sara Rampazzi. Invisible reflections: Leveraging infrared laser reflections to target traffic sign perception. In *NDSS*, 2024.
> >
> > [3] Meixi Zheng, Xuanchen Yan, Zihao Zhu, Hongrui Chen, Baoyuan Wu. BlackboxBench: A Comprehensive Benchmark of Black-box Adversarial Attacks. *IEEE TPAMI*, 2025.

---

> > > ### Comment · Reviewer_1t4h · 2025-08-06
> > >
> > > Thank you for the clarification. My concerns regarding the black-box setting and the stealthiness of the attack have been addressed by the authors’ explanation. However, for Table 2, if the attack is indeed query-based and conducted in a black-box setting, I believe the paper should report the number of queries required to perform the attack. This is a critical metric for evaluating the practicality and efficiency of a black-box approach.
> > >
> > > Regarding Section 4.1, I understand that the goal of this component is to ensure that the perturbation is applied strictly within the traffic sign region. The authors explain that without the region localization step, the perturbation often appears outside the valid physical region. However, my intuition is that perturbations placed far from the target object (e.g., outside the traffic sign) should not significantly influence the detector’s predictions due to the limited receptive field of CNN-based detectors. If spatially distant perturbations can still consistently affect detection outcomes, this would be quite unexpected.

---

> > > > ### Author Response · Authors · 2025-08-07
> > > >
> > > > **We sincerely appreciate that our previous response addressed your concerns regarding the black-box setting and stealthiness. Thank you as well for your valuable suggestions, which will help guide further clarification and improvements in the revision.**
> > > >
> > > > **Q-1. I believe the paper should report the number of queries required to perform the attack. This is a critical metric for evaluating the practicality and efficiency of a black-box approach.**
> > > >
> > > > A-1. We greatly appreciate your constructive feedback. We agree that the number of queries is a crucial metric for evaluating the efficiency of our black-box method. In the PSO function we used, there are three parameters related to the number of queries:
> > > >
> > > > - `n_restarts`: the number of random restarts.
> > > > - `swarmsize`: the number of particles in each PSO run.
> > > > - `maxiter`: the maximum number of iterations in each PSO run.
> > > >
> > > > The number of queries can be calculated as: **Queries** = `n_restarts` × (`swarmsize` × `maxiter`). For our FIPatch, the total number of queries is **1500**, with **`n_restarts = 5`**, **`swarmsize = 10`**, and **`maxiter = 30`**. We will report the number of queries required to execute the attack in the revision. If necessary, we can also test the impact of the number of queries on ASR.
> > > >
> > > > **Q-2. Regarding Section 4.1, I understand that the goal of this component is to ensure that the perturbation is applied strictly within the traffic sign region. The authors explain that without the region localization step, the perturbation often appears outside the valid physical region. However, my intuition is that perturbations placed far from the target object (e.g., outside the traffic sign) should not significantly influence the detector’s predictions due to the limited receptive field of CNN-based detectors. If spatially distant perturbations can still consistently affect detection outcomes, this would be quite unexpected.**
> > > >
> > > > A-2. We apologize for any confusion caused by our previous response. In our previous response, we stated that “if the location of the perturbation is not restricted, the model can still be misled **even** when the perturbation lies outside the traffic sign region”. This highlights that there are some **exceptional cases** where the perturbation located outside the traffic sign region can still successfully mislead the model, rather than suggesting that the perturbation **often appears** outside the sign area.
> > > >
> > > > Your intuition is **correct**. For CNN models with limited receptive fields, perturbations that are far from the target object are indeed unlikely to significantly affect the model’s predictions. In practice, we observed that without the localization step in Section 4.1, perturbations may succeed in the digital domain but fail in the physical world. This is not because the perturbation is placed far from the sign, but due to a **mismatch** between the **location** of the perturbation in the digital and physical environments. Specifically:
> > > >
> > > > - The perturbation lies very close to the TS, but not exactly on it.
> > > > - The center of the perturbation may be within the traffic sign, but parts of it extend beyond the boundary, resulting in an incomplete shape after physical deployment.
> > > >
> > > > This discrepancy arises because, during digital optimization, the perturbation is applied as a complete circular region. However, if any portion of this circle falls outside the actual sign region, it may be physically cut off or distorted when deployed, thereby reducing attack effectiveness.
> > > >
> > > > To prevent this issue, Section 4.1 introduces a masking mechanism that strictly constrains the perturbation within the traffic sign region at each optimization step. This ensures that the generated perturbation fully complies with the shape and placement constraints of the real-world scenario.  As a result, the perturbation remains effective and consistent across both digital and physical environments.

---

> > > > > ### Comment · Reviewer_1t4h · 2025-08-07
> > > > >
> > > > > Thank you for the responses. Most concerns are resolved, but I suggest revising the unclear part for clarity. I will update my score accordingly.

---

> > > > > > ### Author Response · Authors · 2025-08-07
> > > > > >
> > > > > > Thank you for the constructive and valuable comments! We will clarify the unclear part in the revision accordingly.

---

> ### Author Response · Authors · 2025-08-08
>
> We appreciate your positive feedback and are glad that our revisions address your concerns. If there are any remaining issues that need to be resolved, please let us know as we are happy to make further improvements.

---

### Official Review · Reviewer_XS8P · 2025-07-03

**Clarity:** 3
**Significance:** 3
**Originality:** 3
**Rating:** 4
**Confidence:** 4

**Summary:**

This paper proposes FIPatch, a novel physical adversarial attack against traffic sign recognition (TSR) systems using fluorescent ink. Unlike conventional attacks that rely on visible stickers, projections, or acoustic signals, FIPatch leverages the stealthy properties of fluorescent materials, which remain invisible under normal conditions but are activated by ultraviolet (UV) light. The authors design a comprehensive framework to optimize the shape, color, and placement of fluorescent perturbations digitally before physically applying them. FIPatch achieves high attack success rates—over 98% in low-light conditions—and can mislead TSR systems to hide, forge, or misclassify traffic signs. It also bypasses five popular defense mechanisms. Extensive real-world and simulation experiments demonstrate its robustness across various models, lighting conditions, distances, and vehicle speeds.

**Questions:**

see the weakness

**Ethical Concerns:**

["NO or VERY MINOR ethics concerns only"]

**Final Justification:**

Thank the authors for addressing my questions. I will keep my score

**Limitations:**

Yes

**Paper Formatting Concerns:**

the formatting is correct

**Quality:**

3

**Strengths And Weaknesses:**

Strengths:

1. A novel adversarial attack form in the physical world, is verified to be effective for traffic sign recognition.
2. The writing is clear, which makes readers can easily understand the paper.

Weakness:

1. In section 3.2, the authors state that there is no direct access to a victim's vehicle. That means the attackers should perform physical attacks based on the transferability. However, as for the method, I find it seems a query-based attack (i.e., using the PSO algorithm). Thus, evaluating the transferability against the unknown threat models is important and necessary in this setting.

2. In Table 2, the authos evalute five models, for generative attack and hiding attack, they test Yolo and faster-rcnn (object detection models), but for misrecognition attack, they test ResNet, VGG, etc (image classification models), why not test object detection models for misrecognition attack. I mean that, for three forms of attacks, all the five threat models should be test.

3. This paper only verified the proposed method in traffic sign recognition task, how about other object detection tasks, like face recognition, general object detection, etc.

4. The contributions should be extended, thus readers can better understand the novelty in this paper. At least the second contribution should point out what is the specific novelty for the proposed method compared with the current methods.

---

> ### Author Rebuttal · Authors · 2025-07-30
>
> **We greatly appreciate your constructive suggestions, which we believe could be addressed in the next version.**
>
> **Q-1. Evaluating the transferability against the unknown threat models is important and necessary in this setting.**
>
> A-1. We agree that evaluating transferability under a black-box setting is particularly important. In our experiments, we tested the transferability of FIPatch and reported the results in **Table 5 in the Appendix**. As shown in Table 5, the attack success rates (ASRs) are above 85% when transferring to models such as ResNet50, ResNet101, VGG13, and VGG16. The lowest ASR is 69%, which occurs when transferring from Inception v3 to MobileNet v2. Notably, models with similar architectures (e.g., ResNet50 to ResNet101) exhibit high transferability, with ASRs reaching up to 95%. These results demonstrate the strong transferability of our attack across various models. In the next version, we will move these results into the main text.
>
> **Q-2. For three forms of attacks, all the five threat models should be test.**
>
> A-2. We apologize for possible misunderstanding about the experiments. As described in Section 4.3, we define goal-based loss functions according to different attack objectives. For hiding and generative attacks, the loss functions are based on the model’s outputs $Pr(object)$ and $Pr(class)$. Note that $Pr(object)$ is the confidence that an object exists in a given cell. Since image classifiers do not produce this objectness score, it is not feasible to evaluate hiding or generative attacks on classification models. Following your suggestion, we have revised Table 2 to include the ASRs of misrecognition attacks against object detectors. The updated results are shown below:
>
> **Table: The ASR of FIPatch on various models in the physical world under misrecognition attack.**
>
> | Light (Lux) | Frames | Yolov3 | Faster R-CNN | ResNet50 | VGG13  | MobileNet v2 | GoogleNet |
> | :---------: | :----: | :----: | :----------: | :------: | :----: | :----------: | :-------: |
> |     200     |  4374  |  100%  |    98.24%    |   100%   | 99.82% |    98.93%    |   100%    |
> |     500     |  3655  | 98.44% |    94.69%    |  99.35%  | 98.01% |    95.38%    |  97.52%   |
> |    1000     |  4163  | 93.27% |    89.41%    |  94.26%  | 92.58% |    92.15%    |  93.66%   |
> |    2000     |  3719  | 89.43% |    84.73%    |  90.59%  | 86.37% |    85.92%    |  84.03%   |
> |    3000     |  3924  | 85.93% |    77.57%    |  84.12%  | 79.55% |    74.59%    |  76.15%   |
>
> We will clarify this distinction in the revised manuscript and incorporate the updated results into Table 2. Furthermore, we evaluated the performance of FIPatch across different models in our ablation study, which further highlights its effectiveness and robustness.
>
> **Q-3. This paper only verified the proposed method in traffic sign recognition task, how about other object detection tasks, like face recognition, general object detection, etc.**
>
> A-3. We chose traffic sign recognition because it is a highly representative task that involves both classification and detection. It is reasonable to extend our method to other scenarios, such as human recognition in surveillance systems, where an attacker could wear clothing with fluorescent materials. As for face recognition, it presents new challenges due to the complex 3D surface of the human face, and we consider it a promising direction for future work. We will include a discussion on the potential extension to other tasks in the next version.
>
> **Q-4. The contributions should be extended, thus readers can better understand the novelty in this paper. At least the second contribution should point out what is the specific novelty for the proposed method compared with the current methods.**
>
> A-4. We are the first to propose the use of fluorescent ink for physical adversarial attacks. Existing methods can be categorized into those using stickers, light, or acoustic signal. However, sticker-based approaches [1]\[2][3] suffer from poor stealthiness and indiscriminately attack all vehicles once deployed. Visible light-based attacks [4]\[5] make it easy to trace the attacker, while invisible light (e.g., infrared [6]) can be easily countered by infrared filters. In addition, acoustic signal-based attacks [7] can be easily blocked by physical signal protection mechanisms. Unlike existing methods, our approach introduces fluorescent ink as a novel attack medium, which significantly enhances stealth and poses greater challenges for defense mechanisms.
>
> Although fluorescent ink enables invisible and actively triggered features, designing an effective and robust physical adversarial patch remains non-trivial with the following challenges. First, it is challenging to simulate fluorescent effects and determine the optimal choices of massive factors in fluorescent ink, e.g., color, transparency, and size, for achieving high attack effectiveness. Second, fluorescent effects are easily influenced by real-world environments, including surroundings, ambient light, and vehicle distance. Existing methods do not address these challenges, and most require manual annotation of traffic sign locations.
>
> To address the above challenges, we design the FIPatch that leverages the unique properties of fluorescent ink. In more detail, our methodology consists of four key modules. First, we develop a color-edge fusion method to automatically locate traffic signs, enabling precise application of fluorescent ink to the signs, rather than invalid backgrounds. Second, to effectively simulate fluorescent effects, we model fluorescent perturbations on traffic signs by defining the various critical parameters of fluorescent ink, including colors, intensities, and perturbation sizes. Third, we design goal-based and area-based loss functions to achieve high ASRs with minimal perturbations, supporting three attack goals: hiding attack, generative attack, and misclassification attack. Finally, to improve the robustness of FIPatch in the physical world, we present several fluorescence-specific transformation methods that simulate fluorescence perturbations for real-world attacks.
>
> The consistency between digital and physical experimental results further validates the effectiveness of our approach. In the next version, we will extend the contributions to elaborate on our innovations and highlight the differences from existing methods.
>
> **Reference**
>
> [1] Kevin Eykholt, Ivan Evtimov, Earlence Fernandes, Bo Li, Amir Rahmati, Chaowei Xiao, Atul Prakash, Tadayoshi Kohno, and Dawn Song. Robust physical-world attacks on deep learning visual classification. In *CVPR*, 2018.
>
> [2] Dawn Song, Kevin Eykholt, Ivan Evtimov, Earlence Fernandes, Bo Li, Amir Rahmati, Florian Tramer, Atul Prakash, and Tadayoshi Kohno. Physical adversarial examples for object detectors. In *WOOT*, 2018.
>
> [3] Aishan Liu, Xianglong Liu, Jiaxin Fan, Yuqing Ma, Anlan Zhang, Huiyuan Xie, and Dacheng Tao. Perceptual-sensitive gan for generating adversarial patches. In *AAAI*, 2019.
>
> [4] Giulio Lovisotto, Henry Turner, Ivo Sluganovic, Martin Strohmeier, and Ivan Martinovic. SLAP: Improving physical adversarial examples with Short-Lived adversarial perturbations. In *USENIX Security*, 2021.
>
> [5] Ranjie Duan, Xiaofeng Mao, A Kai Qin, Yuefeng Chen, Shaokai Ye, Yuan He, and Yun Yang. Adversarial laser beam: Effective physical-world attack to dnns in a blink. In *CVPR*, 2021.
>
> [6] Takami Sato, Sri Hrushikesh Varma Bhupathiraju, Michael Clifford, Takeshi Sugawara, Qi Alfred Chen, and Sara Rampazzi. Invisible reflections: Leveraging infrared laser reflections to target traffic sign perception. In *NDSS*, 2024.
>
> [7] Wenjun Zhu, Xiaoyu Ji, Yushi Cheng, Shibo Zhang, and Wenyuan Xu. Tpatch: A triggered physical adversarial patch. In *USENIX Security*, 2023.

---

> > ### Comment · Reviewer_XS8P · 2025-08-05
> >
> > Thank the authors for addressing my questions. I will keep my score

---

### Decision · Program_Chairs · 2025-09-17

**Decision:**

Accept (poster)

**Comment:**

This paper proposes a physical attack on traffic sign recognition using fluorescent ink and presents a coherent framework that models fluorescence in the digital domain and translates it into a realizable physical design; its novelty and practical stealth are highly notable. Even under a black-box setting, the work demonstrates high attack success rates in real-world conditions and evasion of multiple defenses, and—after the rebuttal—adds evaluations including ViTs and matched-condition comparisons that appropriately mitigate the main concerns. Remaining issues include streamlining the exposition, clarifying the query budget and optimization procedure, and supplementing discussion of UV operational constraints and transferability, all of which appear readily addressable in the camera-ready. Overall, the originality of the medium and the solid empirical evidence substantiate the contribution to the NeurIPS community, and I recommend acceptance at the upper end of the borderline range.